# Multi-scale phenomena of rotation modified mode-2 internal waves

David Deepwell[1], Marek Stastna[1], and Aaron Coutino[1]

[1]Department of Applied Mathematics, University of Waterloo, Waterloo, Canada

*Correspondence to:* David Deepwell (ddeepwel@uwaterloo.ca)

**Abstract.** We present high resolution, three dimensional simulations of rotation modified mode-2 internal solitary waves at various rotation rates and Schmidt numbers. Rotation is seen to change the internal solitary-like waves observed in the absence of rotation into a leading Kelvin wave followed by Poincaré waves. Mass and energy is found to be advected towards the right-most side wall (for Northern hemisphere rotation), leading to increased amplitude of the leading Kelvin wave and the formation of Kelvin-Helmholtz instabilities on the upper and lower edges of the deformed pycnocline. These fundamentally three dimensional instabilities are localized within a region near the side wall, and intensify in vigour with increasing rotation rate. Secondary Kelvin waves form further behind the wave from either resonance with radiating Poincaré waves or the remnants of the K-H instability. The first of these mechanisms is in accord with published work on mode-1 Kelvin waves, the second is, to the best of our knowledge, novel to the present study. Both types of secondary Kelvin waves form on the same side of the channel as the leading Kelvin wave. Comparisons of equivalent cases with different Schmidt numbers indicate that while adopting a numerically advantageous low Schmidt number results in the correct general characteristics of the Kelvin waves, excessive diffusion of the pycnocline and various density features precludes accurate representation of both the trailing Poincaré wave field and the intensity and duration of the Kelvin-Helmholtz instabilities.

## 1 Introduction

Over recent decades nonlinear internal solitary waves (ISWs) have been the subject of continuing research due, in part, to their common presence in coastal waters (Shroyer et al. (2010), Lamb (2004), and Zhang et al. (2015)) and estuaries (Bourgault and Kelley (2003)), and an expanding set of applications, such as plankton and krill transport (Scotti and Pineda (2004),Cuypers et al. (2010)) or cross-shelf transport (Hosegood and van Haren (2004)). Of particular interest are the effects that rotation has on these waves, since they have been observed to have lifetimes such that this effect is non-negligible (Farmer et al. (2009)). At observation sites such as Knight inset (Klymak and Gregg (2001)) and within the St. Lawrence river (Mertz and Gratton (2013)), side walls may also impact the propagation of these waves. Indeed, classical linear wave theory for rotation modified waves demonstrates that the presence of side walls allows a different type of wave to be created, namely a Kelvin wave.

The dominant laboratory insights on Kelvin waves in a channel geometry come from the experimental work of Maxworthy (1983), and Renouard et al. (1987). These authors performed lab-scale experiments in which mode-2 (Maxworthy (1983)) and mode-1 (Renouard et al. (1987)) Kelvin waves were generated in a rectangular domain. Both authors found that though the wave amplitude increased at the channel wall with increasing rotation rate, the phase speed and shape were comparable

to waves of similar amplitude in the presence of no rotation. The authors also described how the wave amplitude decayed exponentially away from the wall, how the wave front was curved backwards, and how the waves decayed as they propagated away from the generation site due to the generation of inertial waves. Melville et al. (1989) followed this by showing that Poincaré waves of comparable phase speed to a Kelvin wave will naturally resonant with the Kelvin wave, thus causing the curvature of the wave front and the amplitude decay away from the side wall.

In Sánchez-Garrido and Vlasenko (2009) the authors discussed numerical simulations constructed to approximately model the evolution of mode-1 waves in the Strait of Gibraltar. When the latitude was increased to 60 degrees, the authors found clear evidence of a secondary tail of Poincaré waves which trailed the leading Kelvin wave and extracted energy from this wave. The authors also found clear evidence of Mach stems. As this study provides a direct comparison work to our own, its results will be discussed at various points in the following. Observation of mode-1 ISW breaking in a non-rotating environment has been presented by Moum et al. (2003). The authors show that localized increases in stratification (a compression of isopycnals) at the wave crest is followed by interfacial overturning and a breakdown to turbulence.

In the simpler case of a rotating fluid adjusting without the presence of side walls, it has been observed that if dispersive effects are accounted for, a leading solitary wave is created which then breaks down into a nonlinear wave packet as it propagates (Coutino and Stastna (2017)). Additionally, a geostrophic state is created at the site of the initial condition which oscillates at a near inertial frequency and radiates waves. Previous work based on hydrostatic equations (Kuo and Polvani (1997)) had suggested that the nonlinear waves created would steepen and eventually break, however, when dispersive effects are accounted for, breaking does not occur and a nonlinear wave packet is generated instead. This effect has been observed in lab scale experiments using the Coriolis Rotating Platform in Grenoble (Grimshaw et al. (2013)). To numerically model these effects the authors used model equations based on the rotation modified Korteweg-de Vries (KdV) or Ostrovsky equation, which gave qualitatively similar results to the observations. However, these equations do not account for the full nonlinearity and dispersion. This accounts for some of the differences with the results in Coutino and Stastna (2017) where the full stratified Euler equations were used.

The work by Fedorov and Melville (1995) showed that if dispersive effects are neglected (when nonlinearity is large), the Kelvin waves will break. Specifically, the authors found that rotation delays the onset of breaking by 60%. When the breaking occurs, it simultaneously forms across the zone of uniform phase that is normal to the boundary. The increase in nonlinearity is seen to create a dipole structure in the cross-shelf velocities. On a similar note, (Kuo and Polvani (1997)) note that the time of breaking depends on both the rotation rate and the steepness of the initial conditions (see their figure 18 for details). However, it is unclear how a more realistic model with short wave dispersion would modify these results.

There have been a number of studies on Kelvin waves from a model equation approach. Grimshaw (1985) derived a rotation-modified Korteweg-de Vries (rKdV) equation whose transverse structure is that of a linear Kelvin wave. The author also showed that when rotation is weak, (the internal Rossby radius is much larger than the wavelength of the wave) the evolution is described by the rotation-modified Kadomstev-Petviashvili (rKP) equation. This was followed up by Katsis and Akylas (1987) who performed a numerical simulation of these equations and found that the wave amplitude varied exponentially across the

channel and the wave front was curved backwards in agreement with the results seen in Maxworthy (1983) and Renouard et al. (1987).

Rotation modified mode-1 ISWs within a cylindrical geometry have been investigated by Ulloa et al. (2014) and Ulloa et al. (2015) using an immersed boundary numerical method. The authors found that the rotation rate affected the nonlinear steepening which further caused a degeneration of the fundamental Kelvin wave into a solitary-type wave packet. When the Kelvin wave amplitude was large enough, localized turbulent patches were produced by Kelvin wave breaking.

Results on non-rotating mode-2 ISWs, especially with regards to their mass transport capabilities (Deepwell and Stastna (2016); Salloum et al. (2012); Brandt and Shipley (2014); Terez and Knio (1998)), are readily available and there is considerable contact between the experimental and numerical modeling literature. This is exemplified by recent progress on quantifying the effects of displacing the pycnocline center from the mid-depth (Carr et al. (2015); Olsthoorn et al. (2013)). In contrast, mode-2 ISWs in a rotating reference frame have been document experimentally by Maxworthy (1983), but no high resolution numerical simulations that provide concrete examples of phenomena future experimental efforts could aim to document exist. We provide such simulations below, with a focus on the overturning induced by the rotation modified ISW (or Kelvin wave, depending on one's choice of terminology) at the focusing boundary.

Our primary qualitative results that could be confirmed in the laboratory concern the fundamentally three dimensional nature of the shear instability at the edges of the mode-2 wave's core, and the the details of the spatial structure of the span-wise kinetic energy flux. The former could be visualized by a PIV system with a light sheet oriented in the span-wise direction, or along-tank light sheets at varying distances from the focussing wall. The latter could be characterized by the more usual along-tank PIV set up. Moreover, the quantitative results of the Kelvin wave-Poincaré wave resonance and the formation of secondary Kelvin waves in our simulation should provide an easier comparison than field oriented simulations such as those of Sánchez-Garrido and Vlasenko (2009). In terms of the numerical modeling literature, we are interested in exploring how the Schmidt number (or Prandtl number in thermally stratified systems) affects the localized shear instabilities generated near the Kelvin wave crest. This is important since Schmidt numbers representative of salt stratification (Sc $\approx 700$) are presently intractable for numerical simulations on all but the smallest scales, but realistic results may be obtained by choosing a Schmidt number larger than that for a heat stratified system (Sc $\approx 7$) but much smaller than that of salt. It also implies that while field scale simulations like those of Sánchez-Garrido and Vlasenko (2009) may have a similar Rossby number to an experimental study, they cannot have the same viscosity and diffusivity, implying that experimentalists need to carefully assess what aspects of such simulations they may successfully observe in the laboratory.

The remainder of the paper is structured as follows: the set up of the numerical experiments and numerical methods are outlined first, with the results that follow structured to identify fundamental differences between rotating and non-rotating evolution, characterize the three-dimensional structure of near wall overturning, and point out the importance of using a realistic Schmidt number other than Sc $= 1$, which has been consistently used in past literature.

## 1.1 Configuration of Numerical Experiments

We have run a series of direct numerical simulations (DNS) in a setup similar to that of Maxworthy (1983), who employed a gravity intrusion from a lock release in a rotating, rectangular tank to generate mode-2 waves. Since the flow develops from a state of rest the precise definition of the term "Direct Numerical Simulation" from the turbulence literature, namely that grid spacing must be smaller than the Kolmogorov microscale, cannot be directly translated to the present situation. We define DNS in the sense commonly adopted in the stratified flow modeling literature, with Arthur and Fringer (2016) providing a concrete example. These authors state that DNS is a three-dimensional simulation which has a grid spacing which is "within approximately one order of magnitude of the Kolmogorov length scale". The Kolmogorov scale for transitional flows is defined in an ad hoc manner, usually via the explicit calculation of the viscous dissipation rate. The grid scale of our simulations is comparable to this usage since it is within an order of magnitude of the Kolmogorov scale defined from the maximum local dissipation rate. Moreover, our numerical method is spectral in all directions, and hence formally higher order than that used in Arthur and Fringer (2016). The spectral filter used to control aliasing applies only to the largest 30% of wave numbers and leaves the majority untouched, and no subgrid scale model as in Large Eddy Simulation (LES) is used. Based on these considerations, and in the absence of a better term, DNS will be used throughout.

We form the gravity intrusion by releasing a large density perturbation into a quiescent, quasi-two-layer background stratification (figure 1 shows this so–called lock-release configuration). Algebraically, the density field has the form,

$$\rho(x,z) = \rho_0 - \frac{\Delta\rho}{4}\left[\tanh\left(\frac{z-z_0-\eta(x)}{h}\right) + \tanh\left(\frac{z-z_0+\eta(x)}{h}\right)\right], \tag{1}$$

where $z_0$ is the location of the background pycnocline, $h$ is the half-width of the pycnocline, and,

$$\eta(x) = \frac{H_m}{2}\exp\left[-\left(\frac{x}{L_m}\right)^p\right], \tag{2}$$

describes the perturbation in which $H_m$, $L_m$, and $p$ set the height, width, and transition length, respectively. The above specification of stratification leads to a single hyperbolic tangent profile for $x \gg L_m$ and a double hyperbolic tangent profile with an intermediate density layer for $x \ll L_m$. As is typical of studies of mode-2 waves (Davis and Acrivos (1967); Stamp and Jacka (1995); Salloum et al. (2012)), the pycnocline is centred at the mid-depth ($z_0 = L_z/2$). The stratification parameters are listed in table 1. For comparison, the domain size is $L_x \times L_y \times L_z = 6.4\,\text{m} \times 0.4\,\text{m} \times 0.3\,\text{m}$.

**Table 1.** Stratification parameters.

| $H_m$ (m) | $L_m$ (m) | $p$ | $\Delta\rho/\rho_0$ | $z_0$ (m) | $h$ (m) |
|-----------|-----------|-----|---------------------|-----------|---------|
| 0.1 | 0.3 | 8 | 0.02 | 0.15 | 0.02 |

We have completed a suite of numerical simulations at various rotation rates and Schmidt numbers. The rotation rate has been specified using the Coriolis parameter, $f$, which is defined in the usual way as twice the rotation rate. We define our maximum Coriolis parameter as $f_0 = 0.105\,\text{s}^{-1}$ which is comparable to the literature (Grimshaw et al. (2013)) and, in particular, to values achievable in the Coriolis Rotating Platform in Grenoble.

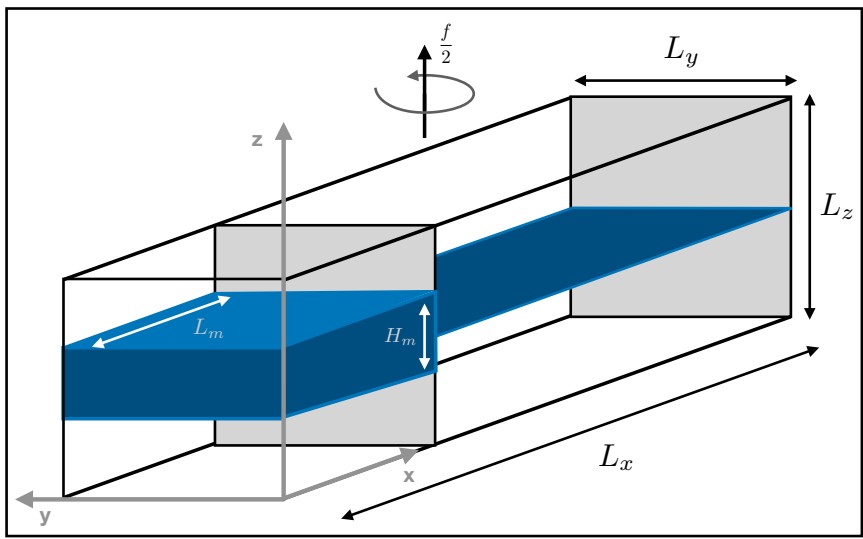

**Figure 1.** Schematic of the numerical domain. The blue region, centred at the mid-depth, corresponds to $\rho = \rho_0$ with heavier fluid below and lighter fluid above. All lengths have the units of meters and the rotation rate has the units of s$^{-1}$.

Following the work of Maxworthy (1983), we characterize the leading wave in regard to its size at the $y = 0$ m wall. In particular, the wave speed, $c_w$, and amplitude, $a_w$, are parametrized at this boundary because it is where they reach their maximal values due to the focusing of mass and energy by rotation. The amplitude is defined as the average maximum upstream displacement of the $\rho(z_0 \pm h)$ isopycnals (for a schematic see figure 2 in Deepwell and Stastna (2016)). A representative value

(table 2) is chosen just prior (in time) to the formation of instabilities. The wave speed, measured as the speed of the location of the maximum displacement, is independent of the rotation rate and has a value larger than the linear long wave speed due to the large amplitude nature of these waves. The amplitude is only weakly dependent on the rotation rate or Rossby number. In general, increasing the initial perturbation, both $L_m$ and $H_m$, leads to the formation of more individual ISWs, larger wave amplitude, and thus higher wave speed. For a comprehensive discussion, consult Deepwell and Stastna (2016);

Salloum et al. (2012); Terez and Knio (1998) or Brandt and Shipley (2014) for more details on the relationship between initial tank parameters and the resultant wave. Discussion about how the ISW is formed from the initial conditions will also be deferred to these articles.

The internal Rossby radius of deformation is defined as $L_c = c_w/f$, the Rossby number as Ro $= c_w/fL_m$, the Reynolds number as Re $= c_w L_m/\nu$, and the Schmidt number as Sc $= \nu/\kappa$. The kinematic viscosity was $\nu = 2 \times 10^{-6}$ m$^2$/s for all cases,

whereas the mass diffusivity, $\kappa$, varied. The parameters and characteristic values for each simulation are presented in table 2. The velocity is scaled by the mode-2, linear, non-rotating, long wave speed, $c_0 = \frac{1}{2}(gh\Delta\rho/\rho_0)^{1/2}$ (Benjamin (1967)). The average rate of change of the amplitude is given by $a' = -\frac{T}{h}\frac{da_w}{dt}$, where $T = L_m/c_0$ is the characteristic time scale.

We briefly contrast our set-up to that of Sánchez-Garrido and Vlasenko (2009). The primary difference is the scale to be modeled. While we seek to model laboratory scale motions, and hence resolve both tank scale and small scale motions, in

**Table 2.** Case parameters and characterizations.

| Case | Sc | $f/f_0$ | $c_w/c_0$ | $a_w/h$ | $a'$ | Re | Ro | $L_c$ (m) |
|------|-----|---------|-----------|---------|------|------|------|-----------|
| 10_0 | 10 | 0 | 1.48 | 1.47 | 1.83 | 6960 | $\infty$ | $\infty$ |
| 10_1/16 | 10 | 1/16 | 1.49 | 1.45 | 2.47 | 7010 | 23.74 | 7.12 |
| 10_1/4 | 10 | 1/4 | 1.48 | 1.49 | 6.02 | 6960 | 5.90 | 1.77 |
| 10_1/2 | 10 | 1/2 | 1.46 | 1.53 | 6.40 | 6870 | 2.91 | 0.87 |
| 10_1 | 10 | 1 | 1.47 | 1.61 | 6.76 | 6900 | 1.46 | 0.44 |
| 4_0 | 4 | 0 | 1.48 | 1.46 | 2.03 | 6930 | $\infty$ | $\infty$ |
| 4_1/2 | 4 | 1/2 | 1.45 | 1.53 | 5.96 | 6820 | 2.89 | 0.87 |
| 4_1 | 4 | 1 | 1.46 | 1.61 | 6.62 | 6870 | 1.45 | 0.44 |
| 1_0 | 1 | 0 | 1.44 | 1.41 | 2.88 | 6770 | $\infty$ | $\infty$ |
| 1_1/2 | 1 | 1/2 | 1.41 | 1.50 | 4.97 | 6640 | 2.81 | 0.84 |
| 1_1 | 1 | 1 | 1.44 | 1.61 | 6.32 | 6780 | 1.43 | 0.43 |

Sánchez-Garrido and Vlasenko (2009) the authors seek to model field scale motions. Hence their ratio of depth to span-wise extent is much smaller than ours. Moreover they model dissipation through an eddy viscosity, while we carry out a Direct Numerical Simulation (DNS). Nevertheless, the ratio of their Rossby radius to span-wise extent is roughly 0.6, hence similar to some of our experiments. Thus many of the large scale motions in the two sets of simulations can be expected to be similar.
5   In particular, figure 2 of Sánchez-Garrido and Vlasenko (2009) makes for a useful comparison to some of our findings.

## 1.2   Numerical Methods

Our numerical model solves the so-called Boussinesq equations of motion on an $f$-plane (Kundu et al. (2012)),

$$\frac{D\boldsymbol{u}}{Dt} + 2\boldsymbol{\Omega} \times \boldsymbol{u} = -\frac{1}{\rho_0}\nabla p + \frac{\rho}{\rho_0}\boldsymbol{g} + \nu\nabla^2\boldsymbol{u}, \tag{3a}$$

$$\nabla \cdot \boldsymbol{u} = 0, \tag{3b}$$

$$\frac{D\rho}{Dt} = \kappa\nabla^2\rho, \tag{3c}$$

where $\boldsymbol{g}$, the gravitational acceleration vector pointing in the negative $z$ direction, $\boldsymbol{\Omega} = f/2\hat{k}$ is the f-plane rotation vector, and other variables have their usual meaning. These equations are presented in dimensional form, while the remainder of this article will use the following non-dimensionalizations,

$$\widetilde{x} = x/L_m, \quad \widetilde{y} = y/L_y, \quad \widetilde{z} = z/L_z, \tag{4a}$$

$$\widetilde{t} = t/T, \tag{4b}$$

$$\widetilde{\rho} = (\rho - \rho_0)/\rho_0, \tag{4c}$$

where the later is called the density anomaly. Scaling time by the rotation rate is inapplicable because the flow is predominantly a Kelvin wave with a propagation speed independent of the rotation rate. Rather, time has been scaled by $T = L_m/c_0$ since the propagation speed of a Kelvin wave is equivalent to the gravity wave speed in the absence of rotation.

The equations used differ from the oceanic situation in that we take the density as a variable to be evolved, whereas in the
5 ocean it is the salinity and temperature that evolve, with density recovered from an equation of state. The nonlinearity of the equation of state leads to a variety of complex phenomena (e.g. salt fingering, cabbeling, the fact that pure water has a density maximum at 4 degrees Centigrade, etc). In the laboratory, density changes are typically imposed by variations in salinity with the temperature held fixed. Our formulation mirrors this situation, though the experimentally observed diffusivity of salt proves too low for inclusion in the numerical simulations.

Numerical simulations were completed using the Spectral Parallel Incompressible Navier-Stokes Solver (SPINS) (Subich et al. (2013)). SPINS is a psuedo-spectral code capable of solving the given problem to a high degree of accuracy in the given geometry. The third-order Adams-Bashforth method with an adaptive time step was used to evolve the flow. Free slip boundary conditions were specified on all walls. This is an unphysical condition compared to the laboratory experiments of Maxworthy (1983), though is standard numerical practice. However, as we will see later, the length scale associated with shear instabilities
is much larger than the boundary layer thickness and hence boundary layer effects will not affect the dominant mechanism for shear production. Besides a minor reduction in the energy within the Kelvin wave, we believe that changing to a no slip boundary condition on the side wall will have minimal impact on the Kelvin wave.

The size of the channel and the grid resolution are listed in Table 3. The stated resolution was sufficient for all but two simulations: case 10_1 and case 4_1. These both had the resolutions in the $x$ and $z$ dimensions doubled to have the total
20 number of points be $N_x \times N_y \times N_z = 4096 \times 256 \times 512$. Grid convergence studies were conducted for these cases because they were the most energetic with the largest and most energetic density overturns. Good agreement was found for the cases with the stated resolution and those with half the resolution. In general, the higher resolutions have been used in this article because of their higher accuracy, though bulk characteristics of the flow computed at the lower resolution remain accurate. For the resolution listed in table 3, the strongly stratified region of the background stratification contains approximately, $2h/\Delta z \approx 33$
points, while the entire stratification has approximately 140 points. Small scale features in the transitional flow typically are a couple centimetres in diameter and contain about 20 points. The applicability of the stated resolution was also found by comparing the grid scale to the Kolmogorov scale which we define using the maximum local energy dissipation rate. In all cases the maximum grid resolution is within an order of magnitude of the Kolmogorov scale. Thus our simulations are well resolved.

**Table 3.** Tank dimensions and numerical resolution.

| $L_x$ (m) | $L_y$ (m) | $L_z$ (m) | $N_x$ | $N_y$ | $N_z$ | $\Delta x$ (mm) | $\Delta y$ (mm) | $\Delta z$ (mm) |
|-----------|-----------|-----------|-------|-------|-------|------------------|------------------|------------------|
| 6.4 | 0.4 | 0.3 | 2048 | 256 | 256 | 3.1 | 1.6 | 1.2 |

## 2 Results: Influence of rotation

We begin by looking at how the ISW is affected by rotation through the Coriolis force. We have chosen the rotation to match that of the Northern Hemisphere which causes objects to be deflected towards the right of their trajectory. In the context of our experiment this leads to span-wise variation in the developing ISW. Maxworthy (1983) found that the wave front became curved as a result of variation of the celerity on wave amplitude. At $\widetilde{y} = 0$ (which we will call the focusing wall) the amplitude and celerity were both larger than at other $y$ values.

We investigate the location of the ISW crests using the scaled, vertically integrated kinetic energy,

$$\xi(x,y,t) = \frac{\int_0^{L_z} \text{KE} \, dz}{\max_{x,y,t} \int_0^{L_z} \text{KE} \, dz} \tag{5}$$

where the kinetic energy is defined in the usual way, $\text{KE} = \frac{1}{2}\rho_0 \left( u^2 + v^2 + w^2 \right)$, and the maximum is over space and time.

The time evolution of $\xi$ displays the bending of the leading internal Kelvin wave and the developing Poincaré waves for case 10_1 (figure 2). Early on (figure 2a), the energy is mostly within the leading wave since insufficient time has passed for

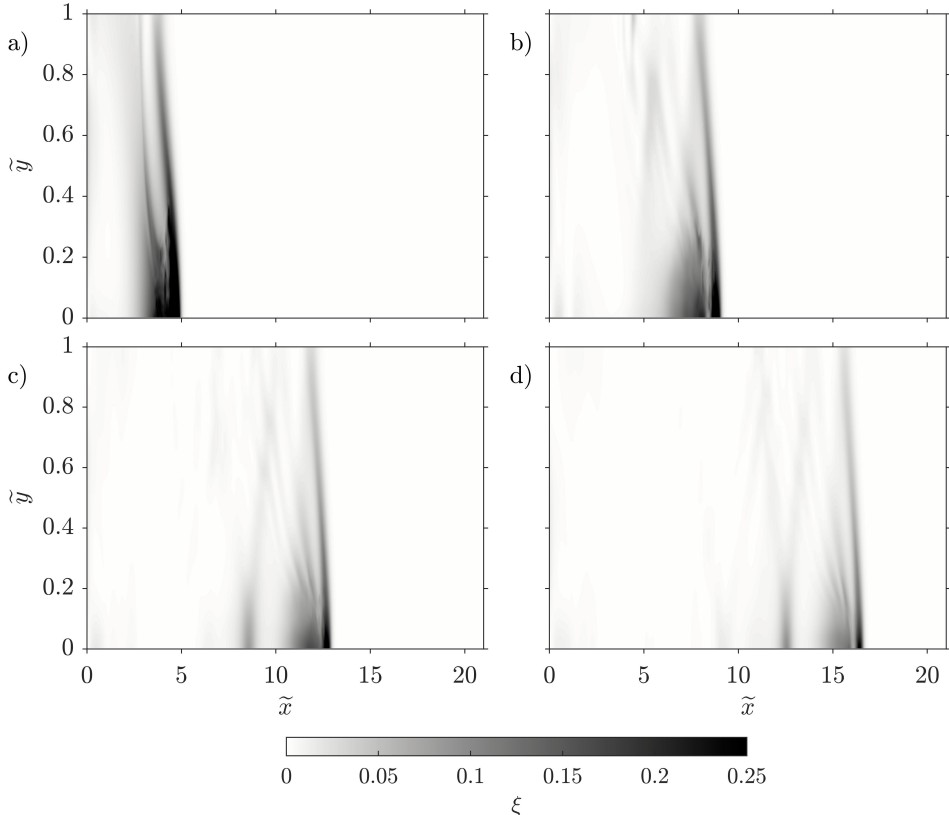

**Figure 2.** The time evolution of the scaled, vertically integrated kinetic energy, $\xi$, for case 10_1 at a) $\widetilde{t} = 2.6$ ($t = 25$ s) b) $\widetilde{t} = 5.2$ ($t = 50$ s) c) $\widetilde{t} = 7.8$ ($t = 75$ s) d) $\widetilde{t} = 10.4$ ($t = 100$ s). Colour axis is saturated at early times to show the wave at later intervals.

radiation to occur. However, at this early time, the wave, which began as a plane wave across the channel, has clearly been affected by the rotation as evidenced by energy being focused towards $\widetilde{y} = 0$. This focusing also resulted in the curvature of the leading Kelvin wave front, a phenomenon which remains evident for the remainder of the simulation.

As time progresses, Poincaré waves form behind the Kelvin wave, as previously described by Sánchez-Garrido and Vlasenko (2009). The Poincaré waves reflect multiple times off both the $\widetilde{y} = 1$ and $\widetilde{y} = 0$ walls. At $\widetilde{t} = 10.4$ ($t = 100$ s) the ratio of total KE on the $\widetilde{y} = 1$ wall to the $\widetilde{y} = 0$ is approximately 0.11 indicating that location of primary activity will be near the focusing wall. The shear also reaches its maximum at $\widetilde{y} = 0$, enabling the onset of dynamic instabilities.

The presence of the Kelvin and Poincaré waves in this context is quite common and is dependent on the rotation rate (figure 3). An increasing rotation rate leads to an increase in the angle that the leading wave makes with the normal of the boundary. The Poincaré waves behave similarly as the rotation rate is varied, however they also have the added complication of reflection and non-linear interaction between waves. When rotation is absent (figure 3a), Poincaré waves are unable to form; instead, a

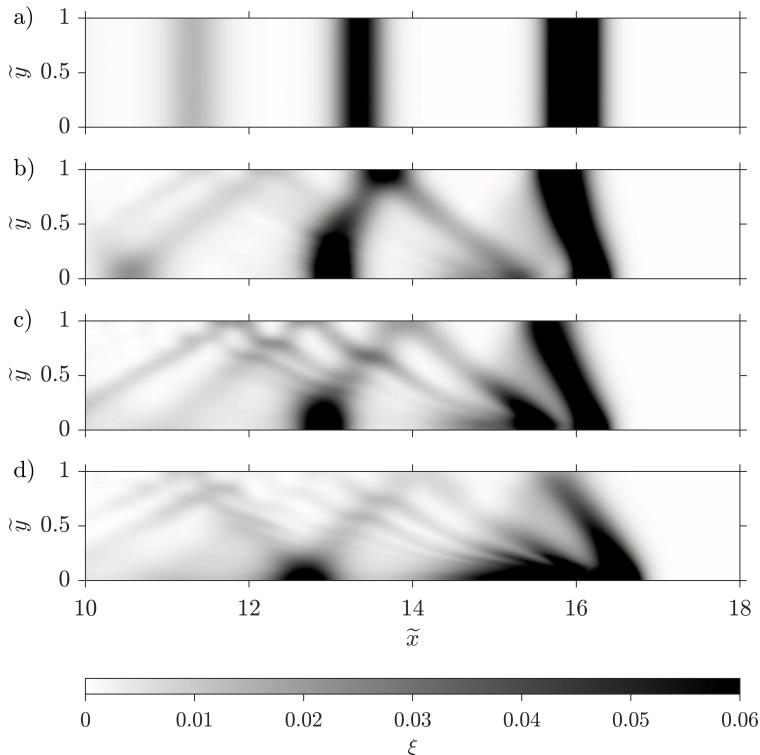

**Figure 3.** Scaled, vertically integrated kinetic energy, $\xi$, at $\widetilde{t} = 10.4$ ($t = 100$ s) for case with a) $f/f_0 = 0$ (Ro= $\infty$) b) $f/f_0 = 1/4$ (Ro= 5.9) c) $f/f_0 = 1/2$ (Ro= 2.91) d) $f/f_0 = 1$ (Ro= 1.46). The scaling is by the maximum over the $f = f_0$ case and the colour axis is saturated to emphasize the Poincaré waves emanating from the focusing region and their reflection off the $\widetilde{y} = 1$ wall. The maximum of the unscaled $\xi$ for each case is a) 0.08 J·m b) 0.12 J·m c) 0.16 J·m d) 0.27 J·m. Though in non-dimensional form, the axes have correct dimensional scaling.

simple train of three planar ISWs of decreasing amplitude and energy are formed. Overall, the spatial distribution of kinetic energy is dominated by the Kelvin wave front and two secondary features near the focusing wall.

It needs to be mentioned that the presence of side walls removes the possibility of a span-wise invariant geostrophic state forming in the collapse region since the presence of walls enforces that flow is in the along tank direction. This means that the release of mass and energy into the ISWs is greater than when no side wall is present. The detailed dynamics of the near field are interesting, but beyond the present manuscript.

A secondary boundary trapped wave also forms in the rotation modified cases (figure 3b-d). The generation mechanism for this wave is fundamentally distinct from the formation of trailing ISWs in the non-rotating case. In discussing this difference, ISW will be used to describe the non-rotating waves only, while Kelvin and Poincaré waves will naturally be understood to relate to the rotation modified waves. To see how this secondary Kelvin wave is formed, figure 4 compares the span-wise average $\xi$ of the non-rotating case (case 10_0, in red) to that of the full $\xi$ for case 10_1 (black pseudocolour). At the early time, $\widetilde{t} = 4.7$ $(t = 45$ s), there is a single leading Kelvin wave and a few radiating Poincaré waves which are just beginning to reflect off the $\widetilde{y} = 1$ wall. The trailing ISWs of the non-rotating case are completely unrelated to the energy distribution of the rotating case. This is clearly evident in figure 4 b) and c) where the second trailing ISW is located where very little energy exists in the rotating case. The third ISW is near where the second Kelvin wave is forming, but this happens to be a coincidence of the choice of the presented time. Later, at $\widetilde{t} = 10.4$ $(t = 100$ s) (figure 3) the third ISW is well behind the second Kelvin wave.

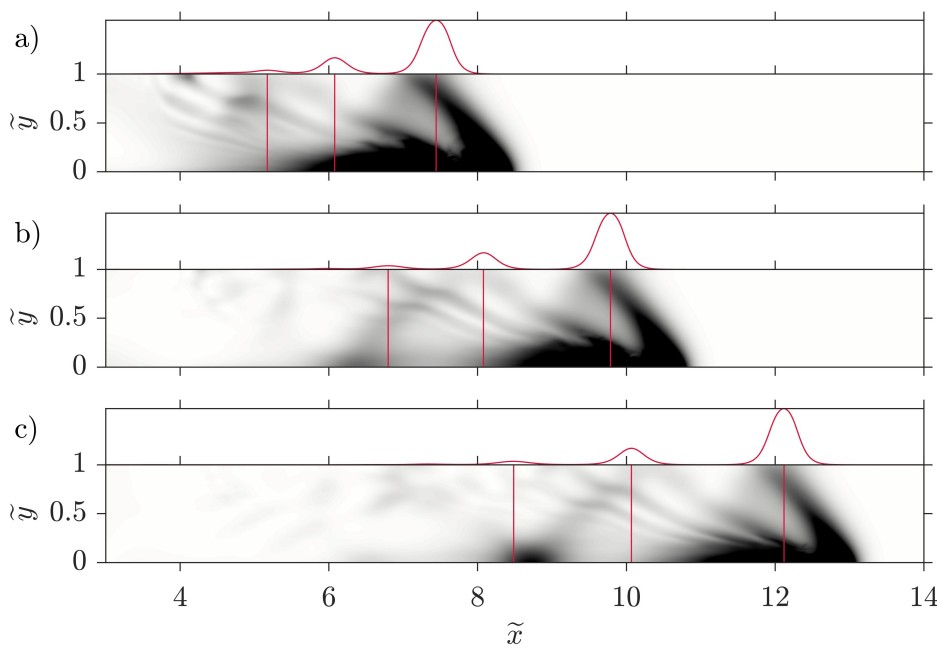

**Figure 4.** Scaled, vertically integrated kinetic energy, $\xi$, for case 10_1 at $t = $ a) $\widetilde{t} = 4.7$ $(t = 45$ s) b) $\widetilde{t} = 6.3$ $(t = 60$ s) c) $\widetilde{t} = 7.8$ $(t = 75$ s). Scaling and colour axis is identical to figure 2. The red plots are the span-wise average $\xi$ profiles for the non-rotating case (Case 10_0), with corresponding maxima locations (vertical lines).

More importantly, at high rotation rates the second Kelvin wave is clearly formed by a resonance with the Poincaré waves at the focused wall, which is directly contrary to how the trailing ISWs form without rotation (they are the excess mass that isn't trapped by the leading wave). This description appears to be valid for rotation rates greater than $f/f_0 = 1/4$, while rates slower than this show the trailing Kelvin wave to match the generation of a trailing ISW in the non-rotating case with additional

perturbations due to weak Poincaré waves. Renouard et al. (1987) describes Poincaré waves being resonantly generated along the side wall while Melville et al. (1989) clearly shows that the curvature of the wave front is due to the combination of a Poincare and Kelvin wave. Our results are in agreement with both of these results, but now applying to mode-2 ISWs.

As described by Sánchez-Garrido and Vlasenko (2009) the Poincaré waves will continue to remove energy from the leading Kelvin wave, which is then deposited in this secondary Kelvin wave. This deposition primarily occurs on the focusing wall,

as opposed to equal deposition on both sides of the channel, because of the broken symmetry caused by the collapse of intermediate fluid on one end of the channel. This means that there is more residual kinetic energy at $\widetilde{y} = 0$ (compared to $\widetilde{y} = 1$), and that the time for the Poincaré wave to resonantly interact with the Kelvin wave is greater at $\widetilde{y} = 0$ since both are traveling in the same direction. Once the secondary Kelvin wave is fully developed, this resonance can be considered analogous to a Mach stem in that the energy builds up on the boundary enough that the reflection no longer occurs at the boundary, and

moves instead some distance away from the wall.

As energy is drained from the leading Kelvin wave and deposited into the secondary wave, the secondary wave will eventually become more energetic than the first resulting in an eventual overtaking. Our simulations do not show this feature since our channel is not long enough and we are focused on the shorter time scales associated with the energetics of the leading Kelvin wave. See Sánchez-Garrido and Vlasenko (2009) for a description of overtaking. Unfortunately, we predict that the overtaking

is likely to remain undetected in a laboratory experiment with rectangular geometry because of the considerable time, and thus length of channel, required for the energy transfer to occur.

The differentiation of whether a wave is a Kelvin wave or a Poincaré wave is made difficult because of the non-linearity associated with the large amplitude of these waves. The classical linear theory, as presented in standard textbooks (Vallis (2006); Kundu et al. (2012)) describes a Kelvin wave as one which has no span-wise velocity, and where the wave crest does not curve

in the span-wise direction. A simple check shows that the leading and radiating waves have significant span-wise velocities which would indicate that they are not Kelvin waves (in the classical sense). However, we choose to label the radiating waves as Poincaré waves and the leading wave as a Kelvin wave because both fit all other descriptions of the particular wave type.

We just saw that the radiating Poincaré waves resonate to form a secondary Kelvin wave. Since this secondary wave is separated from the chaotic leading wave, it could possibly fit better in the description of a classical Kelvin wave. Figure 5

presents a test for this by plotting the vertically integrated span-wise squared velocity (in dark) and contours of the total vertically integrated KE (in red). The secondary wave (at $\widetilde{x} \approx 12.5$) does indeed show no span-wise velocity or wave crest curvature, and thus our description of it as a Kelvin wave is valid. We also notice that the region directly trailing the wave also has little to no span-wise velocity. Looking back at figure 3b and c it becomes clear that there are Kelvin waves directly trailing the leading wave at weaker rotation rates (the highest rotation rate requires more time to settle into this description).

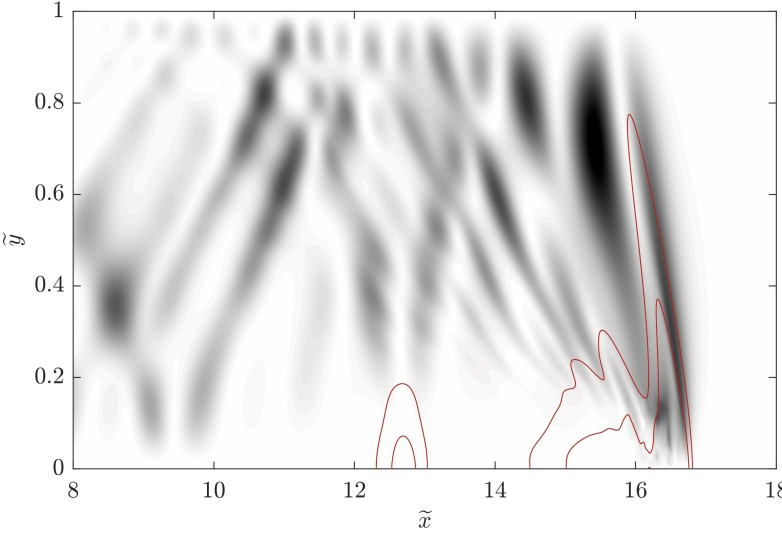

**Figure 5.** Vertically integrated $v^2$ at $\widetilde{t} = 10.4$ ($t = 100$ s) for case 10_1. Red contours are level curves of the vertically integrated KE.

This Kelvin wave is distinct from the previously described secondary wave. In this case they are formed out of the remains of the excess focused mass along $\widetilde{y} = 0$. To distinguish this Kelvin wave from the leading wave, we call it the Kelvin wave tail.

The dynamics seen thus far are fundamentally different than in the case without the side walls. As described in Coutino and Stastna (2017), without the walls the Poincaré waves steepen and eventually breakdown into a non-linear wave packet as dispersive effects take over. The addition of side walls appears to slow the large-scale breakdown of the leading wave. Rather, this energy is moved towards the side wall, and is then radiated leeward into trailing Poincaré waves at a slower pace.

The advection of kinetic energy within the Kelvin and Poincaré waves is key to understanding how the distribution of this energy is influenced by the side wall. The span-wise flux of KE (figure 6) reveals some interesting features. Of primary interest is that the leading Kelvin wave induces both positive and negative span-wise flux of KE. Close to the wall (figure 6c), the front of the wave has advection away from the wall, which then becomes a stronger, more localized return of kinetic energy flux towards the wall within the downstream portion of the leading wave. Further from the side-wall (figure 6a), the KE flux is weaker by nearly an order of magnitude and the structure has changed so that the leading wave contains more positive flux.

The secondary Kelvin wave (near $\widetilde{x} = 12.75$ m) has much weaker KE flux even though the waves are of comparable amplitude. The leading wave still contains residual energy from the K-H instabilities while the secondary wave receives energy from upstream. To within 3%, the same KE flux is advected towards and away from the focusing wall in this secondary Kelvin wave.

For comparison purposes, we have completed the same simulation in two dimensions while allowing transverse flow to be coupled to horizontal motion through the Coriolis force (i.e. a two-and-a-half dimensional model). Since there exists no side wall, the radiated ISWs are smaller because much of the energy remains within the geostrophic state. Regardless of the waves

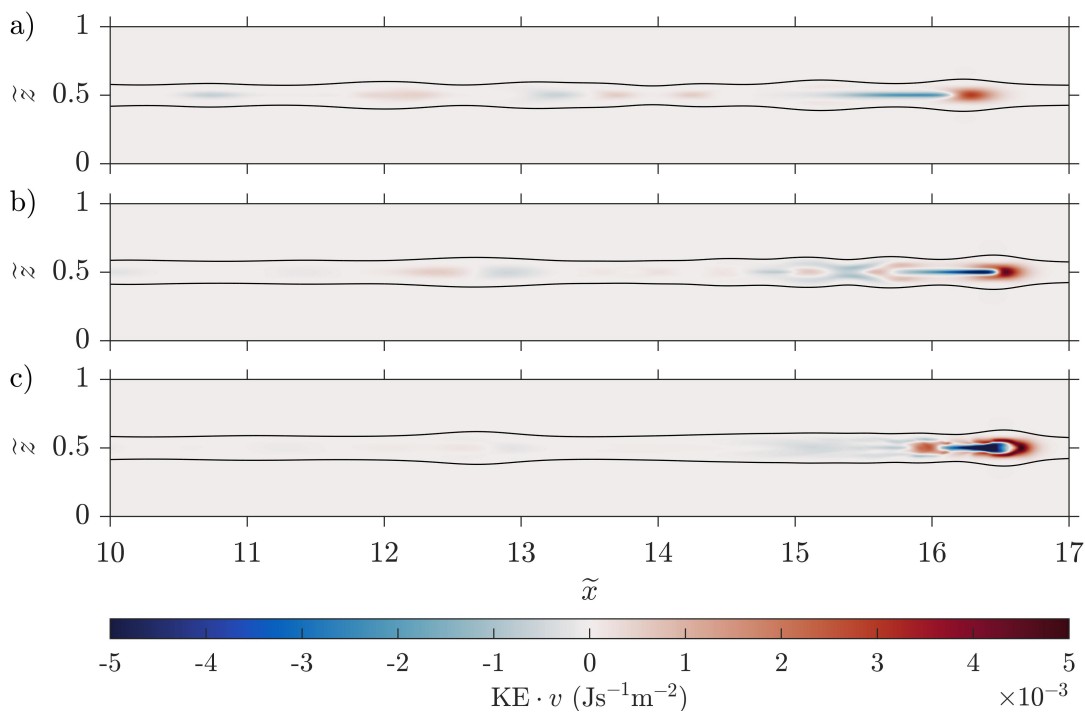

**Figure 6.** Span-wise kinetic energy flux density for case 10_1 at $\widetilde{t} = 10.4$ ($t = 100$ s) and $\widetilde{y} =$ a) 1/2 b) 1/4 c) 1/8.

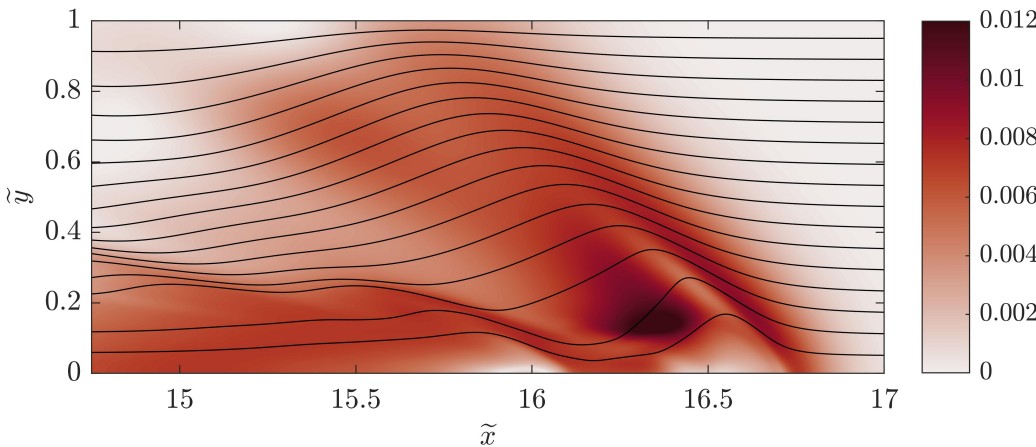

**Figure 7.** Horizontal kinetic energy flux density for case 10_1 at $\widetilde{t} = 10.4$ ($t = 100$ s) and $\widetilde{z} = 0.5$. Streamlines show direction of KE flux in reference frame moving with the leading wave (i.e. flow is right to left). Colour axis shows magnitude of KE flux.

being smaller and thus traveling slower, the KE flux in the leading wave of the 2D case (not shown) is different from that of the 3D case with sidewalls, especially near the wall. Further from the side walls, the two become more similar yet remain distinct in the magnitudes and distributions of the KE flux.

The span-wise variation in the KE flux along the mid-depth ($\widetilde{z} = 0.5$ m) provides a different way to visualize the KE flux away and towards the focusing wall. Figure 7 displays both the magnitude of the flux (colour) and direction (streamlines with flow going from right to left) in a reference frame moving with the wave. The strongest KE flux is directed towards the wall in the aft of the Kelvin wave. We hypothesize that this KE flux is the cause of wave breaking in the Kelvin wave (the details of which we present in the next section). The kinetic energy near the wall then leaves the Kelvin wave and travels along the wall until forming the Kelvin wave tail some time later.

## 3    Results: Details of wave breaking

Now that the description of the global wave field has been presented, we move onto the localized behaviour of the leading Kelvin wave at the focusing wall. As described by Maxworthy (1983), the dominant instability takes the form of a pair of Kelvin-Helmholtz (K-H) billows at the crests of the leading Kelvin wave at the upper and lower extrema of the wave (figure 8a), or, alternatively, at the edges of the pycnocline.

In the highest rotation rate (case 10_1) the instability is confined to within approximately ten centimetres (a quarter of the channel width) of the focusing wall (figure 8b-d). As the rotation rate decreases, the extent of the instability decreases. Table 4 lists the length scale, $l^*$, of the initial billow and the non-dimensional duration of the shear instability, $t^*$. Except in case 10_1, the instabilities stop within the duration of the simulation. For case_1 the size of the K-H billows decreases significantly near the end of the simulation. It is worth repeating that higher rotation rates (smaller Rossby numbers) lead to larger and longer lasting shear instabilities within the crests of the Kelvin wave.

**Table 4.** Length and time scales associated with localized shear instabilities.

| Case | $l^*$ (m) | $t^*$ |
|------|------|------|
| 10_0 | 0 | 0 |
| 10_1/16 | 0.02 | 1.35 |
| 10_1/4 | 0.05 | 1.88 |
| 10_1/2 | 0.07 | 7.93 |
| 10_1 | 0.10 | >12 |

For all cases, the Kelvin-Helmholtz billows simultaneously form as pairs with a vortex above and below the pycnocline. Furthermore, except during the early energetic K-H formation, these billows remain synchronous even during the process of being broken down. Comparison to the observations of Moum et al. (2003) is evident as upstream isopycnals are compressed along the wave crest followed by the K-H billows. These K-H billows can also be considered analogous to a von Karman vortex street. In a reference frame moving with the wave, the background flow is directed around the wave, much like it is around a cylinder in von Karman's classical experiment. The analogy is limited since the Kelvin wave core is not a solid, and the core bends backward away from the wall due to the rotation (and hence isn't a cylinder). Moreover, the instability is a shear instability, as opposed to a boundary layer separation.

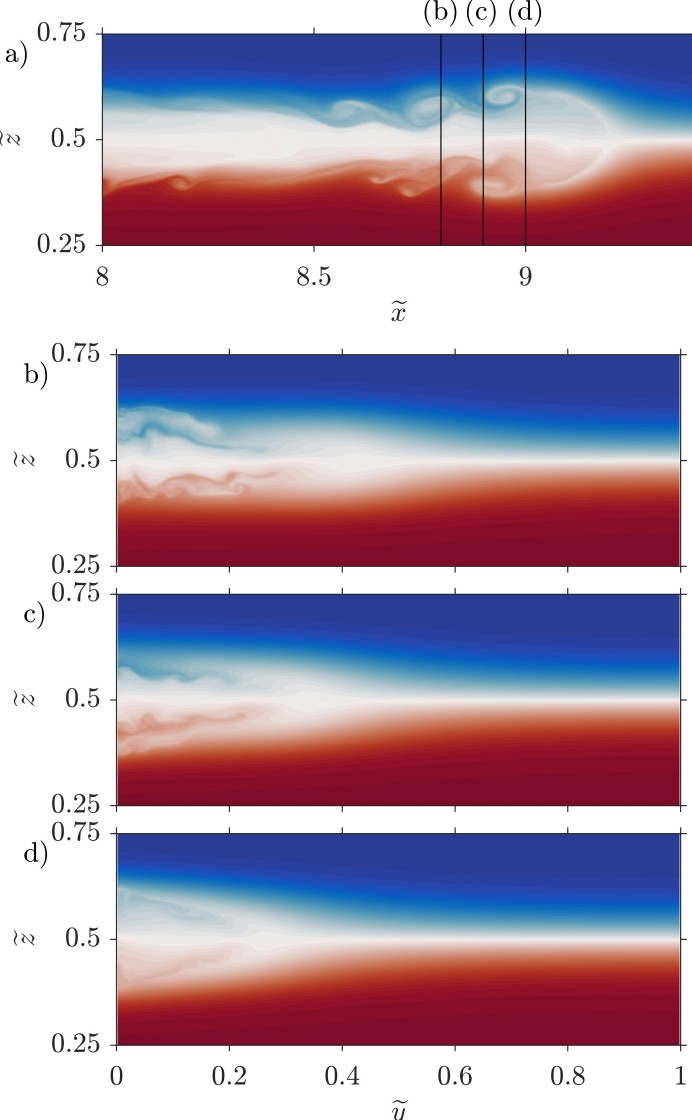

**Figure 8.** Density anomaly, $\widetilde{\rho}$, at $\widetilde{t} = 5.2$ ($t = 50$ s) and a) $\widetilde{y} = 0$ b) $\widetilde{x} = 8.8$ c) $\widetilde{x} = 8.9$ d) $\widetilde{x} = 9$ for case 10_1.

At the wave crest, the wave has the expected exponential decay (figure 8d). Further behind, a description that is purely in the $\widetilde{x} - \widetilde{z}$ plane is no longer valid since the wave front curves backwards, as discussed in the previous section. However, the instabilities remain trapped along the side wall (figure 8b and c), while the wave itself, which created the instabilities, remains stable further away. In the density field, the instabilities are recognized as interleaving layers of lighter and denser
5  fluid associated with the roll-up of the K-H billows. The span-wise extent remains largely unchanged as the vortices leave the leading wave. This means that rotation causes mixing and turbulence to occur at a preferred location, namely on the focusing

wall. Should the geometry, or environmental forcing (e.g. flow over a sill in a fjord) cause Kelvin waves to be generated at a specific location, this would indicate that one side of the channel would experience more mixing. The aforementioned fjords, as well as narrow lakes would be particularly susceptible to this.

We find that changes in the rotation rate (i.e. the strength of the Coriolis force) influence the intensity of the K-H billows (figure 9). Though the initial available energy is the same in all cases, the higher rotation rates lead to higher localized kinetic energy density. At early times (first column of figure 9) the amplitude of the leading waves are all comparable, but have varying levels of stability. The cases with increasing rotation are far more energetic. Fundamentally, these vortices are formed from stratified shear instabilities commonly associated with larger amplitude waves (Brandt and Shipley (2014)). This leads us to suggest that a coastally trapped ISW has a lower minimum amplitude threshold for instability generation compared to a ISW

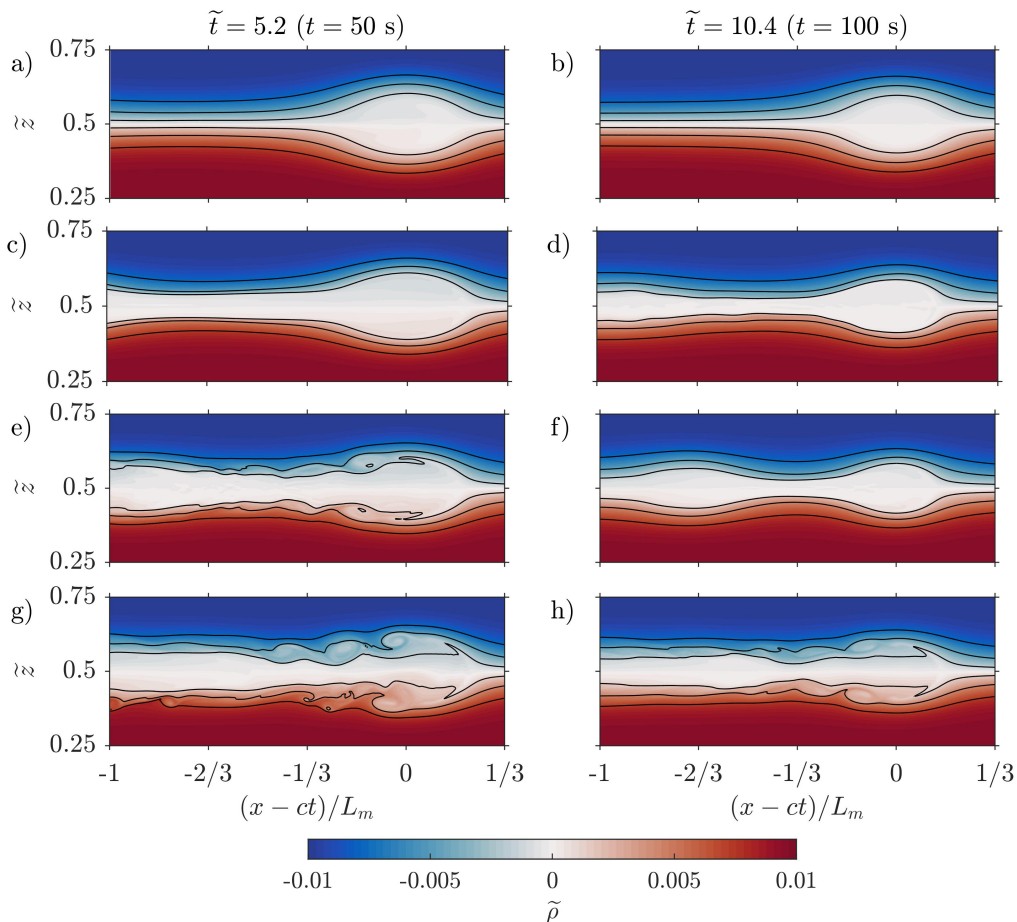

**Figure 9.** Density anomaly, $\widetilde{\rho}$, at $\widetilde{y} = 0$ for $f/f_0 =$ a,b) 0 c,d) 1/4 e,f) 1/2 g,h) 1. Black contours are equispaced isopycnals between $\rho(z_0 - h)$ and $\rho(z_0 + h)$. All cases have Sc = 10.

away from boundaries. The precise reason for the lower minimum amplitude remains an open problem, but it likely has to do with the mass and energy flux towards the focusing region discussed in the previous section.

As time progresses, the wave sheds energy and mass in the shear instabilities until reaching a critical amplitude, after which the wave remains stable, but continues to decay because of the lossy behaviour of the core region of large amplitude mode-2 ISWs (Deepwell and Stastna (2016)). At $\widetilde{t} = 10.4$ ($t = 100$ s), only the highest rotation rate still produces shear instabilities. In comparison, the non-rotating case is laminar and time-invariant, apart from small dissipative effects for all times. The weaker rotation rate cases do not exhibit instabilities at later times, but rather the mass from the shed vortices has formed the Kelvin wave tail. This wave is also immediately apparent from figure 3 which shows the kinetic energy within this wave directly behind the leading Kelvin wave. Over time the lower rotation rates show that this Kelvin wave tail obtains energy from the leading wave and will eventually overtake it. At the highest rotation rate, the shear instability remains active for the entire simulation which makes it difficult for the Kelvin wave tail to form.

The emergence of shear instabilities is correlated with the rotation rate. That is, a higher rotation rate is associated with an earlier shear instability. The higher rotation causes greater fluid to be directed towards the focusing wall which leads to a greater initial amplitude which creates favourable conditions for shear instabilities. Though the initial amplitude is correlated with rotation rate (figure 10), the three fastest rotations result in similar wave amplitudes as the experiment continues. This leads to the interesting question of whether Kelvin waves have a stability restriction based on their amplitudes, though the form of their creation here (as a collapse of a span-wise invariant perturbation) greatly impacts the dynamics, especially with the K-H billows. Though the Richardson number is not applicable in the upper and lower layers where there is no vertical density variation, within the wave core the Richardson number drops below 1/4 around the edge of the mode-2 bulge while the K-H billows form.

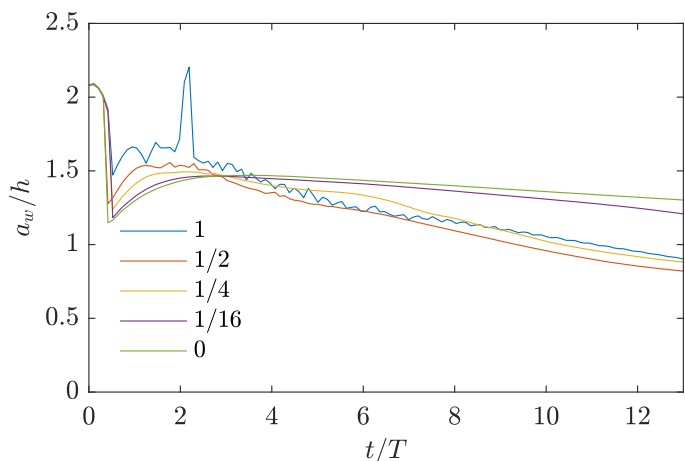

**Figure 10.** Wave amplitude as a function of time for different rotation rates ($f/f_0$ in legend). All cases have Sc $= 10$.

The wave amplitude decay rate, $a' = -\frac{T}{h}\frac{da_w}{dt}$, increases with rotation rate (table 2, and figure 10). At larger rotation rates this is a result of a greater initial amplitude, while slower rates create far less K-H billows and the associated loss of mass. At $\tilde{t} = 5.2$ ($t = 50$ s), the kinetic energy within the leading wave increases with rotation rate (first column of figure 11). The kinetic energy also becomes localized in space at the K-H billows. At later times, The kinetic energy has decreased most substantially in the higher rotation rates because of the shear instabilities.

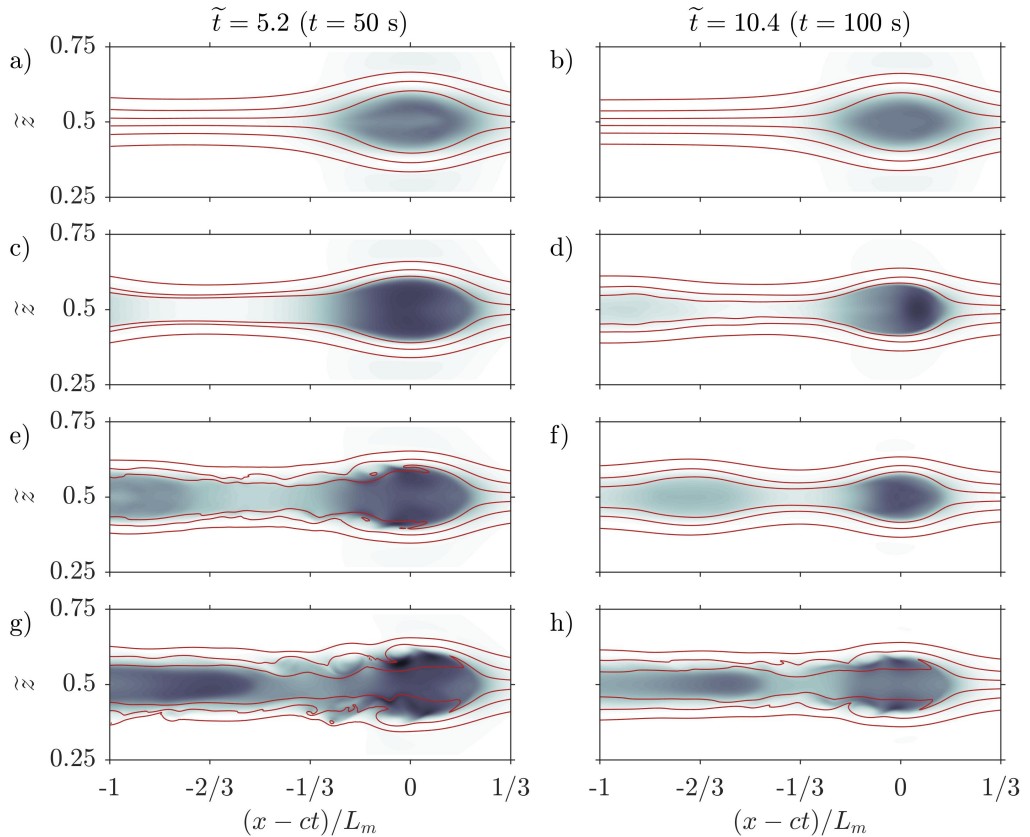

**Figure 11.** Kinetic energy density at at $\tilde{y} = 0$ for $f/f_0 =$ a,b) 0 c,d) 1/2 e,f) 1. Red contours are the same as in figure 9. All cases have $Sc = 10$.

Figure 12 shows the kinetic energy slices in the $\tilde{y} - \tilde{z}$ plane at the crest of the Kelvin wave (ISW in the non-rotating case). The case of no rotation has an essentially span-wise invariant wave front, whereas the rotation breaks the symmetry by curving the wave front. Along the cross-section of maximum amplitude this appears as an exponential decay of the wave amplitude. Essentially all of the kinetic energy resides within the characteristic isopycnals, $\rho(z_0 \pm h)$. Furthermore, the density overturns resulting from the instabilities are correlated with the locations of maximum KE.

Over time, due to the radiation of energy into trailing waves, the leading Kelvin wave reduces in amplitude, span-wise width, and KE. The time dependent nature of the width, and thus the exponential decay, results in an increased localization of KE

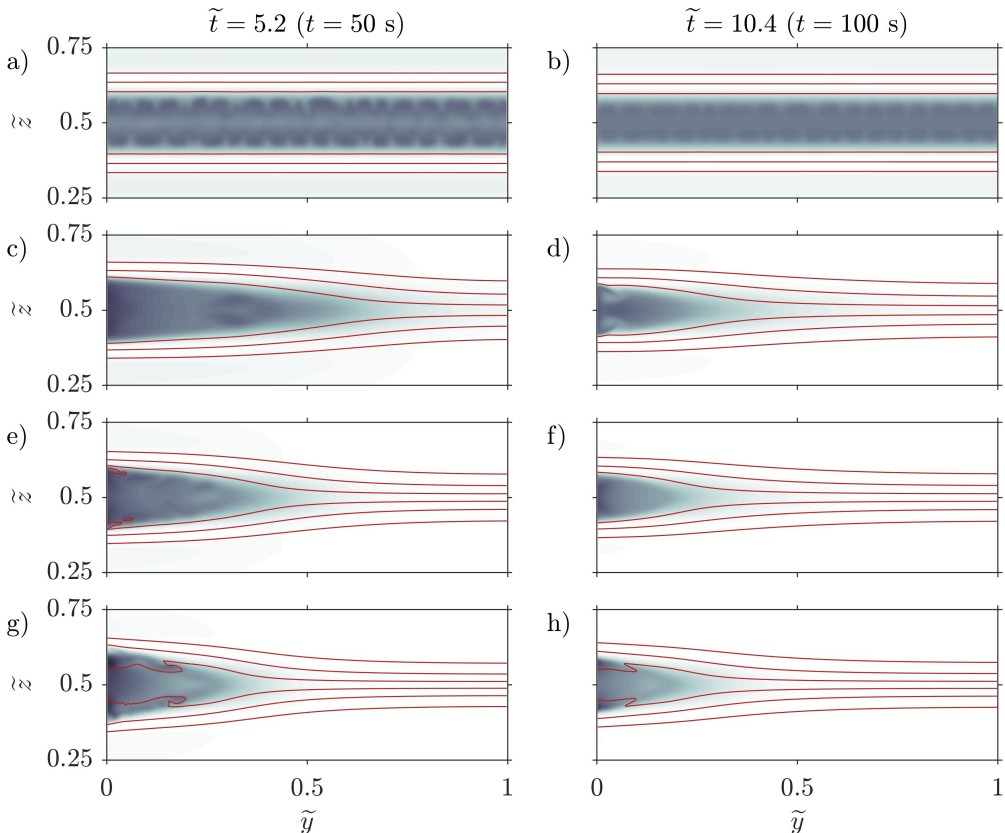

**Figure 12.** Kinetic energy density at the location of maximum amplitude for $f/f_0 =$ a,b) 0 c,d) 1/4 e,f) 1/2 g,h) 1. Red contours are the same as in figure 9. All cases have Sc $= 10$.

along the focusing wall (seen very clearly in the second row of figure 12). This is true along the cross-section of maximum amplitude where the decaying Kelvin wave applies. In the curved portion of the leading wave, this coincides with a simple reduction of the width of the curved wavefront.

Maxworthy (1983) found that the measured Rossby radius of deformation, $L_M$, (that is, the distance from the wall for the amplitude to drop by a factor of $e$) was a factor of two smaller than the calculated Rossby radius, $L_c$. Only the highest rotation rate has $L_M$ smaller than the width of the channel for us to make a comparison. Case 10_1 has the factor being close to 2.6, which is comparable. This does not appear to be the case at lower rotation rates since these cases have their amplitudes decay over the entire channel width even though the Rossby radius increases by up to a factor for 4. This could be due to the narrowness of the channel compared to the Rossby radius or more likely a difference in measurement technique. Maxworthy (1983) made his estimate based on the projection of the wave field onto the $\widetilde{y} - \widetilde{z}$ plane. This would lead to a weaker decay rate since the wave curves backwards away from the cross-section with a fixed $\tilde{x}$ at which our estimate is made. In other words the method of Maxworthy (1983) integrates in the along tank direction, while we choose a particular stream-wise location for

our estimate. The width of the channel will, however, adjust the reflection of the Poincare waves and thus the resonance of the secondary Kelvin wave, but this will have no impact on the leading wave.

## 4  Results: Schmidt number dependence

The shear instabilities and associated dynamics are fundamentally small scale behaviour which are damped by viscosity and smeared by diffusion, both of which are determined by the properties of the fluid, namely the molecular diffusivity and the viscosity. Experimentally, the diffusivity is fixed by the choice of stratifying solute. Physical values of a salt stratified experiment, which are typical for experiments of this type, give a Schmidt number of approximately 700. Since direct numerical simulations at these values are unattainable due to the resolution required, we provide here a short description of the impact that various Schmidt numbers have on the results presented thus far. Of importance is measuring the change in the shear instabilities by varying the Schmidt number.

For longer simulations, such as the ones conducted here, the pycnocline will diffuse causing the waves to propagate in a slightly different stratification near the end of the simulation compared to the beginning. Smaller Schmidt numbers have greater diffusion causing a greater impact (figure 13a). The $Sc = 1$ case had the pycnocline grow by 70% while the $Sc = 10$ case only grew by 10%. The background stratification at the end of the experiment is noticeably different (figure 13b).

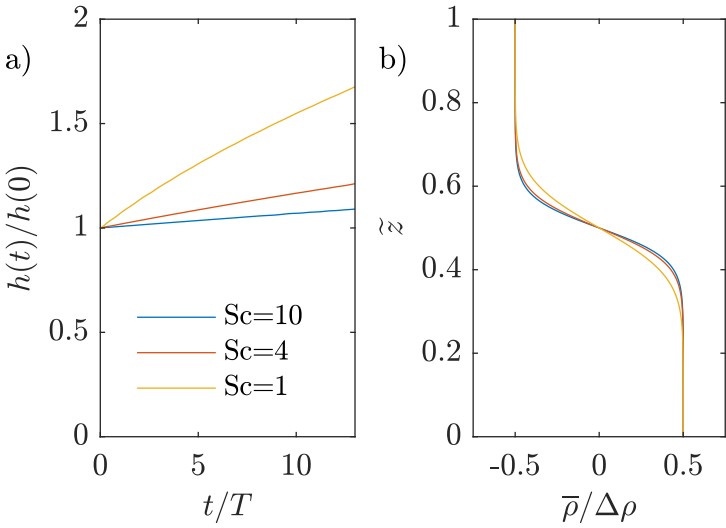

**Figure 13.** a) Pycnocline half-width as a function of time for different Schmidt numbers. b) The background stratification at $\tilde{t} = 12.5$ ($t = 120$ s).

We find that the wave amplitude is unaffected when compared at various Schmidt numbers. Thus, the general features characterizing the wave are fairly similar at $\tilde{t} = 5.2$ ($t = 50$ s) (figure 14). The details, however, are significant enough to warrant comment. All Schmidt numbers experience the shedding of Kelvin-Helmholtz billows, but in the lowest Schmidt

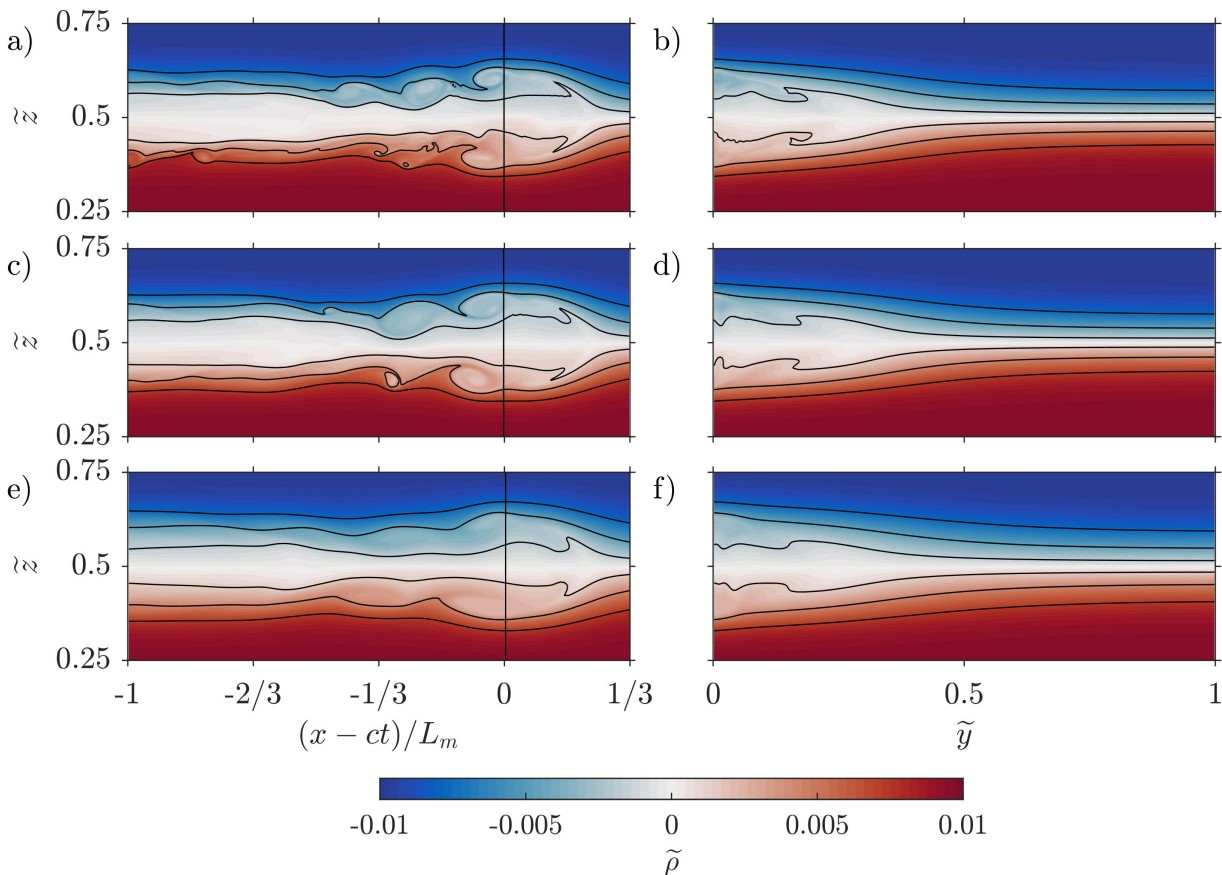

**Figure 14.** Density anomaly, $\widetilde{\rho}$, at $\widetilde{t} = 5.2$ ($t = 50$ s) with cross-sections at $\widetilde{y} = 0$ (first column) and at the location of maximum amplitude (second column) for Sc = a,b) 10 c,d) 4 e,f) 1. Black contours are the same as in 9. Vertical, black line denotes location of the cross-section shown in the right column. All cases have $f/f_0 = 1$.

number case formation ceases at around $\widetilde{t} = 11.7$ ($t = 112$ s) while the other two cases create billows for the duration of the simulation. The higher diffusivity in the Sc $= 1$ case acts to hinder the production of shear instabilities by quickly diffusing them as they form.

    The span-wise profile of the wave also shows the difference (right column of figure 14). The highest Schmidt number case

5  has overturning which is not present in the lowest Schmidt number case. These features are secondary to the over-all behaviour. For example, the total kinetic energy is comparable between all cases at a given rotation rate but remains consistently weaker for lower Schmidt number. The greatest separation between the Sc=1 and Sc=10 case is 13%. As this ratio is only expected to grow as Sc increases, the difference between typical simulations of Sc=1 are bound to misrepresent the smaller dynamical features of an equivalent physical experiment which is typically salt stratified with a Schmidt number around 700. As a result

10  of this, the dynamics of a physical experiment will look like figure 14a,b with substantially finer details. A thermally stratified

experiment, however, will be well represented by figure 14a,b. We are unaware of any experimental work on mode-2 internal waves with stratification achieved by varying temperature.

In the $f/f_0 = 1/2$ rotation rate cases (comparison not shown) the differences are more obvious. The density field does not clearly show the Kelvin wave tail in the Sc=1 case. Rather, the wave is of such small amplitude that it is nearly indistinguishable compared to the pycnocline width. Only by looking at the vertically integrated KE does the wave appear, but is of considerably smaller magnitude. The fanning structure seen in figure 4 also goes from at least two branches in case 10_1/2 to one in case 1_1/2. This results in a different Poincaré wave field in the majority of the domain.

## 5    Conclusions

We have performed a series of numerical experiments of a lock–release configuration exploring the effects that rotation and side walls have on the evolution of mode-2 ISWs. When the stratification has a single pycnocline form, naturally occurring mode-2 waves are quite likely to exhibit regions of overturning and hence our configuration is ideal for exploring the combined effects of rotation and instability. Matching with the results of Maxworthy (1983), we observe that the leading wavefront becomes curved by rotation and that the dominant instability takes the form of two K-H billow trains at the crest of the wave, above and below the pycnocline center. By modifying the rotation rate we observe that as the rate of rotation increases, the instabilities become more energetic. This appears to be due to increased focusing of mass and the kinetic energy density along the focusing wall, especially in the early period of adjustment after the lock is released. Since these types of shear instabilities are most commonly formed by large amplitude waves, this suggests that Kelvin waves in a channel have a lower minimum amplitude threshold for instability generation and thus will have smaller amplitude waves compared to internal solitary waves in the open ocean.

These instabilities cause the leading wave to lose energy and thus higher rotation rates result in a faster decay in wave amplitude than cases with a lower rotation rate. The increased instability generation along the side wall also serves to create an asymmetry in the extent of mixing across the width of the tank. This effect could be observed in fjords or narrow lakes by making a careful comparison of mixing levels and wave amplitudes across the channel.

The high level of mass and kinetic energy along the focusing wall also resulted in the radiation of Poincaré waves, as previously described by Sánchez-Garrido and Vlasenko (2009). The Poincaré waves reflect off the opposing wall before returning to resonantly generate the secondary Kelvin wave along the focusing wall. When compared against an equivalent non-rotating case we found that this generation mechanism (through resonance) at higher rotation rates was completely different from the creation of secondary ISWs in a non-rotating reference frame. We referred to this wave as the secondary Kelvin wave. In addition to this, a Kelvin wave tail formed from the remains of the focused mass and energy along the wall. During propagation, we observed that the leading Kelvin wave appeared to lose energy to the both of these Kelvin waves, each of which became more energetic and thus faster. Eventually, these trailing Kelvin waves will overtake and surpass the leading wave, though we did not witness this phenomena. The results observed are fundamentally different than those seen without side walls. In the case without the side walls the leading wave energy is deposited into the trailing waves through dispersion eventually forming

a wave packet. This packet has no span-wise variation. With side walls, the trailing waves are a fan of Poincaré waves that exhibit a complex interference pattern and hence have a span-wise structure. Furthermore, the energy of the leading wave is not continuously being lost to the primary trailing wave and hence this wave lives significantly longer, compared to the non-rotating case.

The results presented above suggest two clear avenues for future work. One avenue would focus on the rotation modified instability region. While in the above, clear evidence of transitional behaviour was presented, it is unclear to what extent a truly turbulent state was achieved. This is because trapping by the leading Kelvin wave is incomplete and turbulence may lose energy due to a spreading in space. Moreover, while the resolution of the numerical simulations was excellent for the full domain, a study that is focused on turbulent transition could optimize the domain and stratification parameters. For example, the domain could be shortened and the initial perturbation increased in size to increase the wave amplitude. While we have speculated that no-slip sidewalls would not fundamentally alter the shear instability, this remains to be confirmed by using a clustered, Chebyshev grid in the span-wise direction. A more likely mechanism to alter the shear instability is to raise the pycnocline such that the layers are of unequal depths to better approximate an oceanographic stratification.

A second possible avenue for future work would explore the effect of the span-wise extent. Figure 15 of Sánchez-Garrido and Vlasenko (2009) suggests that for wider domains, mode-1 Kelvin waves yield Poincaré wave trains whose focusing yields Mach stems on the far wall. We did not observe this phenomenon in our simulations, but it is possible that our span-wise extent was simply not large enough to achieve this. Of course, an experimental realization of our simulations would provide both a test of our results, and suggestions for future numerical studies that are most relevant to the experimentalist.

*Competing interests.* No competing interests are known.

*Acknowledgements.* Time dependent simulations were completed on the high-performance computer cluster Shared Hierarchical Academic Research Computing Network (SHARCNET, www.sharcnet.ca). DD and AC were supported by an Ontario Graduate Scholarships while MS was supported by an NSERC Discovery Grant RGPIN-311844-37157. Valuable comments by two anonymous reviewers are gratefully acknowledged.

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
