# Peer review of "Multi-scale phenomena of rotation modified mode-2 internal waves"

_Nonlinear Processes in Geophysics, 2017_

## Referee Comment (RC1) · Anonymous Referee #1 · 4 Jan 2018

The paper reports a series of numerical experiments on evolution of second mode ISWs propagated in a laboratory-scale wave tank. The focus of this study is on the effect of rotation that converts the propagated ISW into a series of Kelvin and Poincare internal waves. The wave dynamics described here is entirely expected, although to my knowledge the model results considered here are novel and look realistic, so in my opinion they ought to be presented to wider audience. There are also a few points where the paper can be strengthen, and they are as follows:

Page 3, Line 11: I do not think the authors conducted DNS as they claim in line 11. For such kind of simulations the grid step should be at the level of the Kolmogorov's scale, but there are no details on both in the text. And what about numerical viscosity? With quite a coarse grid it can be several order higher than the molecular viscosity, $2*10^{-6}$

mˆ2/sec, as claimed in the paper.

Page 4, line 8: The authors take the first-mode phase speed as the velocity scale, although the whole model set-up is for the 2-nd mode experiments. Does this make sense?

I'm not sure I understand the meaning of two concepts, cw and aw. They are introduced in line 2 on page 4 in a very general way, without clear explanation how do they relate to the model set-up. However, they appear in table 2 as input parameters. What is the link of these values with the tank experiment parameters (size scales, stratification parameters, rotation, etc)? And why the wave speed, as it is introduced on page 4, is larger than the fastest mode 1 wave speed C0? It seems to me the authors did not pay much attention how their paper will be accepted by the readers.

Relatively minor, but important: The presented on page 5 system is not the NS-system as stated. Please, be careful defining the total water density and its perturbations. Secondly, the temperature, salinity and the EoS are the constituents of the NS-type system, but not the density perturbation (find also a mistake in the first eqn.)

I'm not sure why do the author change the Sc number? They call it the Schmidt number (why not the Prandtl number?, but never mind) and vary it from 1 to 10. This does not make any sense if the authors conduct their experiments for the laminar-size grid. The viscosity and diffusivity coefficients are constant at the Kolmogorov scale level (laminar!!), so why the authors considered their ten times variation (Table 2)? What is the idea behind that? Finally, what is the spatial grid resolution after all? Looking at Table 3 I can see it is at the level of 10-3 m (i.e. 1mm), which is small, but does not tell me whether this is small enough for replication of the laboratory-scale experiments and the background mixing. Maybe yes, but the text in its present state is not convincing enough for me.

No details are provided how the initial ISW was created. Figure 1 does show the initial installation, and I can believe that in the vertically symmetrical case the leading ISW is

a second mode wave, but it really takes time to form in the front of the wave field. Is 6.4m tank long enough to form it? When the rotation has been switched on? Right in the beginning of the experiment? What is the idea of all these experiments? I would accept the method of initial wave formation and initiation of the rotation after that to learn the effect of rotation, but all the details must be explained. I'm really confused without the correct setting of the experiment conditions. Lines 15-25 on page 6 do not bring any clearance on this point.

---

## Referee Comment (RC2) · Anonymous Referee #2 · 5 Jan 2018

**General Comments**

The paper shows exciting numerical results of the evolution of a second-mode internal solitary-type wave in a rotating and stratified channel. The manuscript describes well the degeneration of a leading internal solitary wave both in rotating and non-rotating environments. In the case of rotating environments, the authors find that the initial internal solitary-type wave evolves into a Kelvin-like wave which degenerates in a train of secondary Kelvin waves and Poincaré waves. The authors examine the emergence of K-H shear instabilities as a function of two controlling parameters, the Rossby number, and the Schmidt number, finding that smaller Rossby numbers are associated with an intensification of the shear instabilities, while small Schmidt numbers are associated

with weak shear instabilities.

I think the manuscript fits within the journal scopes, it is well structured, concise and it has clear figures. However, I think the article could examine more in-depth the topics addressed in, especially in the matter of the transition to turbulence, and have a clear message and scopes for the readers. Before the paper is recommended for publication, the authors should address the following comments:

**Mayor Comments**

- There is no comparison between the numerical results and field observations, or attempts to suggests how and where the processes discussed in this manuscripts could be observed in nature. I think is relevant to motivate the readers with some realistic applications of the paper's outcomes. For instance, the authors might compare their numerical experiments in the absence of rotation with the laboratory experiments performed by Carr et al. (2015), since they have collaborated on recent work (Deepwell et al. 2017).

- The authors do not discuss the implications of the free-slip boundary conditions adopted in their numerical experiments; this is the case of Maxworthy (1983). For instance, how does the growth of shear instabilities near no-slip walls would change the results?

- The authors explore only one background stratification aspect ratio $h_1/h_2$, $h$, with different wave amplitudes. I wonder why they did not explore other background stratification. I would expect that in nature an upper layer thinner than a deep layer, or vice-versa. It might be interesting to explore how the asymmetry in layer thicknesses would change the growth and structure of K-H like instabilities.

- The title of the article suggests that the focus of the manuscript is the transition

to turbulence driven by second vertical mode internal waves in a rotating environment, but the paper is more general than this title. The paper examines the macro- and micro-scale processes driven by the degeneration of second vertical mode solitary waves in a rotating and bounded stratified flow. I would suggest the authors think of a different title.

**Specific Comments**

1. In the introduction, one would also cite the work by Moum et al. (2003) or more recent observations by Zhang & Alford (2015), and for motivations, for instance, Cuypers et al. (2010).

2. On page 2, between 20 and 25, I would mention the results obtained by Melville et al. (1989).

3. How did you estimate that a single turbulent patch has a vertical extent of about 40 cm, where are the 25 points coming from? Vertical grid points? Are the authors making a reference to Ulloa et al. (2015)'s paper?

4. Subsections 1.1 and 1.2: Why equations are not enumerated? Please, enumerate them.

5. Page 3, paragraph 20: It seems obvious, but the authors could write that the domain is $L_x \times L_y \times L_z = 6.4\,\mathrm{m} \times 0.4\,\mathrm{m} \times 0.3\,\mathrm{m}$, or something like that.

6. Table 1: Missing units of $z_0$ and $h$; both variable should be defined and schematised in Figure 1.

7. Page 4, paragraph 10: ' ... we *attempt* to carry out a Direct Numerical Simulation (DNS) ... '. Concerning this sentence, what is the Kolmogorov scale of the numerical experiments? What is the Ozmidov scale? If authors are solving the Kolmogorov scale, please take the word 'attempt' out of the sentence, otherwise explain further.

8. Page 4, paragraph 10: Did Sanchez-Garrido & Vlasenko (2009) define in the same way the Rossby number? It seems that their Rossby number is almost twice smaller than the smaller $Ro$ considered in this work.

9. Table 2 is referenced in page 4; so far there is no clear explanation about how $c_w$ and $a_w$ were estimated. The authors explained these parameters were parametrised on the wall, $y = 0$ m, and the amplitude was defined before the formation of instabilities. Was the emergence of interfacial instabilities defined by visual inspection of the isopycnals? How was then estimated the phase speed, $c_w$? This matter requires a bit of further explanation.

10. Subsection 1.2: I would suggest rewriting the description of the governing equations. I would suggest something like 'our numerical model solve the Boussinesq equations of motion on an $f$-plane ...'. 'stratified Navier-Stokes equations' sounds a bit unusual.

11. Page 5, paragraph 15: Is SPINS the acronym given to the numerical solver or is a general pseudo-spectral method?

12. Page 6, paragraph 5: What type of computational resources were used to perform the numerical experiments (machine, number of cores, computational time)?

13. Page 6, paragraph 5: How many vertical grid points does the numerical experiment use to solve the peaks of the square of the buoyancy frequency, $N^2$, on the initial solitary wave? I would include this information to show that the density

transitions are well solved. For instance, the vertical length-scale of the pycno-cline thickness in the wavefront, $\sim h$, is solved by around 17 and 36 grid points each vertical grid resolution, respectively.

14. Section 2. The authors start refereeing the work of Maxworthy (1983). Similarly to ?, I would compute the relationship between wave amplitude, $A_w$, phase celerity, $c_w$, and the dependence with the controlling parameter, the Rossby number (and later with the Schmid number). These results can be discussed along with Figure 3.

15. Equation 1. The definition of $\xi$ does not allow a quantitative comparison between the numerical experiments. I would recommend using a scaling that allows comparison.

16. Page 6, paragraph 25. Where does the shear reach its maximum? at $\tilde{y} = 0$?

17. Figure 3: I would include the Rossby number for each experiment to show the background rotation environment along with the resulting wave dynamics.

18. Section 2 (Page 7, paragraph 5) How did the authors compute the energy partition between Kelvin and Poincaré waves? The authors may compute spatial spectra as a function of time to quantify the energy contained in the spanwise and streamwise axes.

19. Page 8, paragraph 10. How is this Kelvin-Poincaré wave resonance compared with the one studied by Melville et al. (1989), and the observed by Renouard et al. (1993). It seems that the train of solitary waves obtained by Ulloa et al. (2015) also converges to azimuthal secondary Kelvin waves.

20. Page 9, paragraph 10: Is there any experimental evidence that a secondary Kelvin wave becomes more energetic and overtakes the leading Kelvin wave? What is the relevance of this phenomenon? This might be possible to show, for

instance, by using a circular basin such as those used by Wake et al. (2005) and Ulloa et al. (2014). However, this would be a different problem. I think the authors should emphasise the relevance and implications of having secondary Kelvin waves in the system.

21. Page 12, paragraph 5: Is there any relationship between the internal Rossby radius of deformation and the length-scale where shear instabilities were found in the spanwise direction? The authors state that this region is confined to a quarter of the transversal length-scale, $L_z$. However, did they observe any change in terms of the Rossby radii?

22. Page 12, paragraph 10: Why did the oscillatory K-H like billows disappear once the vertical resolution was increased? Any further thought? What was the most critical wavenumber in both cases, when $n_z = 256$, and $n_z = 512$. The authors could perform a stability analysis to understand the nature of the instabilities and the growth rate.

23. Page 12, paragraph 10: Regarding the von-Karman like vortices. I would think that this kind of instabilities is possible to have a physical sense. There is a localised solitary wave propagating through a mid-layer in an initially quiescent stratified fluid. If we move on the leading solitary wave on the near-wall zone, the solitary wave feels there is a streamwise flow in the opposite direction. This scenario leads to a shear flow that could be similar than the one observed in the generation of von-Karman street vortices. Could you please give a look at the vertical velocity profile along the core of the leading wave?

24. Page 13, paragraph 5: Is there any clue of baroclinic-like instabilities? Have the authors observed the vorticity field in the $x - y$ plane?

25. Page 14, paragraph 10: Did the authors compute the local gradient Richardson numbers on the leading solitary wave zone?

26. Page 16, paragraph 10: Do the authors think that the channel width plays any role in the train of solitary-type waves structure and the shear instabilities growth? If so, please comment.

27. Page 17, paragraph 10: What grid resolution would be required to solve the Batchelor scale in a numerical experiment with $Sc \approx 700$? How far are we to solve this problem?

28. Page 18, paragraph 10: 'Since the stratification has broadened in this case, the Richardson number has increased leaving the unstable region for shear instabilities to form.' Please, explain better this sentence.

29. Results: Schmidt number dependence. What is the message from this section? In a thermally stratified fluid ($Pr \approx 7$) we could expect dynamics such as those shown in Figure 14(a,b).

30. There is a problem with the references, the doi link is duplicated.

31. Abstract: I do not understand what does the last sentence mean: '*Comparisons of equivalent cases with different Schmidt numbers indicate that while low Schmidt number results in the correct general characteristics of the modified ISWs, it does not correctly predict the trailing Poincaré wave field or the intensity and duration of the K-H instabilities*'. What is not possible to predict at low Schmidt number (guessing the authors refer to Sc = 1)? How are the authors predicting the trailing Poincaré wave field and the intensity and duration of K-H instabilities?
* * *

---

## Author Comment (AC1) · 30 Jan 2018

We thank the Reviewers for their comments. We have responded to all comments, and where applicable have added in their feedback into the article. Reviewer comments are in bold font, responses are in normal font.

**Page 3, Line 11: I do not think the authors conducted DNS as they claim in line 11. For such kind of simulations the grid step should be at the level of the Kolmogorov's scale, but there are no details on both in the text. And what about numerical viscosity? With quite a coarse grid it can be several order higher than the molecular viscosity, $2 \times 10^{-6} \mathrm{m^2}/sec$, as claimed in the paper.**

The text has been modified to include the following paragraph addressing this issue:

[Figure]

"We have run a series of direct numerical simulations (DNS) in a setup similar to that of Maxworthy (1983), who employed a gravity intrusion from a lock release in a rotating, rectangular tank to generate mode-2 waves. Since the flow develops from a state of rest the precise definition of the term "Direct Numerical Simulation" from the turbulence literature, namely that grid spacing must be smaller than the Kolmogorov microscale, cannot be directly translated to the present situation. We define DNS in the sense commonly adopted in the stratified flow modeling literature, with Arthur and Fringer (2016) providing a concrete example. These authors state that DNS is a three-dimensional simulation which has a grid spacing which is "within approximately one order of magnitude of the Kolmogorov length scale". The Kolmogorov scale for transitional flows is defined in an ad hoc manner, usually via the explicit calculation of the viscous dissipation rate. The grid scale of our simulations is comparable to this usage since it is within an order of magnitude of the Kolmogorov scale defined from the maximum local dissipation rate. Moreover, our numerical method is spectral, and hence higher order than that used in Arthur and Fringer (2016). The spectral filter used control aliasing applies only to the largest 30% of wavenumbers and leaves the majority untouched, and no subgrid scale model as in Large Eddy Simulation (LES) is used. In the absence of a better term, DNS will be used throughout."

**Page 4, line 8: The authors take the first-mode phase speed as the velocity scale, although the whole model set-up is for the 2nd mode experiments. Does this make sense?**

The presented velocity scale is the correct mode-2 phase speed rather than the mode-1 phase speed. We have clarified this point in the manuscript to remove further confusion.

**I'm not sure I understand the meaning of two concepts, $c_w$ and $a_w$. They are introduced in line 2 on page 4 in a very general way, without clear explanation how do they relate to the model set-up. However, they appear in table 2 as input parameters. What is the link of these values with the tank experiment parameters**

**(size scales, stratification parameters, rotation, etc)?**

$c_w$ and $a_w$ are characterizations (not initial parameters) of the resultant mode-2 ISW which describe the measured amplitude and wave speed along the focusing wall. $a_w$ is related to the depth of the initial perturbation, $H_m$, while $c_w$ is amplitude dependent. $a_w$ and $c_w$ have maxima along the focusing wall and have their values decrease along a normal to this surface. The choice of $a_w$ and $c_w$ was made following Maxworthy (1982). We have included a short discussion and included additional references on how these parameters are associated with tank experiment parameters.

**And why the wave speed, as it is introduced on page 4, is larger than the fastest mode 1 wave speed $c_0$? It seems to me the authors did not pay much attention how their paper will be accepted by the readers.**

The wave speed is faster than the linear, mode-2, long wave speed because these waves are highly non-linear and have a strong amplitude dependence. This is a result for finite amplitude mode-2 waves (see Terez and Knio (1998), Brandt and Shipley (2014), or Salloum et al. (2012)). For clarity, we have added a comment about the amplitude dependence of the wave speed which leads to this larger value.

**Relatively minor, but important: The presented on page 5 system is not the NS-system as stated.**

We have fixed this.

**Please, be careful defining the total water density and its perturbations. Secondly, the temperature, salinity and the EoS are the constituents of the NS-type system, but not the density perturbation (find also a mistake in the first eqn.)**

We have explained the set of governing equations used and their relation to the set of equations believed to apply to the oceanic situation: "The equations used differ from the oceanic situation in that we take the density as a variable to be evolved, where as in the ocean it is the salinity and temperature that evolve, with density recovered

from an equation of state. The nonlinearity of the equation of state leads to a variety of complex phenomena (e.g. salt fingering, cabbeling, the fact that pure water has a density maximum at 4 degrees Centigrade, etc). In the laboratory, density changes are typically imposed by variations in salinity with the temperature held fixed. Our formulation mirrors this situation, though the experimentally observed diffusivity of salt proves too low for inclusion in the numerical simulations."

The set of equations we use is standard throughout the literature, but the distinction with the oceanic situation is worth discussion.

**I'm not sure why do the author change the Sc number? They call it the Schmidt number (why not the Prandtl number?, but never mind) and vary it from 1 to 10. This does not make any sense if the authors conduct their experiments for the laminar-size grid. The viscosity and diffusivity coefficients are constant at the Kolmogorov scale level (laminar!!), so why the authors considered their ten times variation (Table 2)? What is the idea behind that?**

We have rewritten the introduction so as to have the reason for varying the Schmidt number be clear at the outset. An extract from this paragraph reads: "In terms of the numerical modeling literature, we are interested in exploring how the Schmidt number (or Prandtl number in thermally stratified systems) affects the localized shear instabilities generated near the Kelvin wave crest. This is important since Schmidt numbers representative of salt stratification (Sc $\approx 700$) are presently intractable for numerical simulations on all but the smallest scales, but realistic results may be obtained by choosing a Schmidt number larger than that for a heat stratified system (Sc $\approx 7$) but much smaller than that of salt. It also implies that while field scale simulations like those of Sanchez-Garrido and Vlasenko (2009) may have a similar Rossby number to an experimental study, they cannot have the same viscosity and diffusivity, implying that experimentalists need to carefully assess what aspects of such simulations they may successfully observe in the laboratory."

**Finally, what is the spatial grid resolution after all? Looking at Table 3 I can see it is at the level of $10^{-3}$ m (i.e. 1mm), which is small, but does not tell me whether this is small enough for replication of the laboratory-scale experiments and the background mixing. Maybe yes, but the text in its present state is not convincing enough for me.**

We have added "For the resolution listed in table 3, the strongly stratified region of the background stratification contains approximately, $2h/\Delta z \approx 33$ points, while the entire stratification has approximately 140 points. Small scale features in the transitional flow typically are a couple centimetres in diameter and contain about 20 points. The applicability of the stated resolution was also found by comparing the grid scale to the Kolmogorov scale which we define using the maximum local energy dissipation rate. In all cases the maximum grid resolution is within an order of magnitude of the Kolmogorov scale. Thus, our simulations are well resolved."

**No details are provided how the initial ISW was created. Figure 1 does show the initial installation, and I can believe that in the vertically symmetrical case the leading ISW is a second mode wave, but it really takes time to form in the front of the wave field. Is 6.4m tank long enough to form it? When the rotation has been switched on? Right in the beginning of the experiment? What is the idea of all these experiments? I would accept the method of initial wave formation and initiation of the rotation after that to learn the effect of rotation, but all the details must be explained. I'm really confused without the correct setting of the experiment conditions. Lines 15-25 on page 6 do not bring any clearance on this point.**

It was assumed that a reader either has a prior knowledge of tank scale mode-2 experiments or is prepared to follow some of the references provided, but obviously this needed to be cleared up. Based on the reviewer comment, we have ensured that some discussion of the lock-release generation method is provided and that sufficient references have now been given which explain in detail the process by which the ISW is

formed. For the initial conditions of the manuscript, the resultant mode-2 ISW is essentially formed within a meter of the collapse region. This is sufficiently quick and leaves the majority of the tank to be used as the domain for the rotation affected ISW. The rotation is present from the moment the simulation begins (as would be the case in a laboratory realization of our numerical set up). Lines 15-25 on page 6 describe a way to measure the location of the wave front in the x-y plane by tracking the kinetic energy. This is used to show the span-wise variation in the ISW rather than the vertical variation which is normally done.

---

## Author Comment (AC2) · 30 Jan 2018

We thank the Reviewers for their comments. We have attempted to respond to all comments. In the case of Reviewer 2, some very insightful comments require computational resources/computation time beyond the one-month revision period and we have taken these as suggestion for future work. Reviewer comments are in bold font, responses are in normal font.

**1 Major comments**

**There is no comparison between the numerical results and field observa-**

**tions, or attempts to suggests how and where the processes discussed in this manuscripts could be observed in nature. I think is relevant to motivate the readers with some realistic applications of the paper's outcomes. For instance, the authors might compare their numerical experiments in the absence of rotation with the laboratory experiments performed by Carr et al. (2015), since they have collaborated on recent work (Deepwell et al. 2017).**

The reviewer is correct that we were not very clear on this point. We have added a paragraph to the introduction that outlines where we see our study in terms of field, experimental and numerical approaches:

"Results on non-rotating mode-2 ISWs, especially with regards to their mass transport capabilities (Deepwell and Stastna (2016), Salloum et al (2012), Brandt and Shipley (2014), Terez and Knio (1998)), are readily available and there is considerable contact between the experimental and numerical modeling literature. This is exemplified by recent progress on quantifying the effects of displacing the pycnocline center from the mid-depth (Carr et al. (2015), Olsthoorn et al. (2013)). In contrast, mode-2 ISWs in a rotating reference frame have been document experimentally by Maxworthy (1983), but no high resolution numerical simulations that provide concrete examples of phenomena future experimental efforts could aim to document exist. We provide such simulations below, with a focus on the overturning induced by the rotation modified ISW (or Kelvin wave, depending on one's choice of terminology) at the focusing boundary.

Our primary qualitative results that could be confirmed in the laboratory concern the fundamentally three dimensional nature of the shear instability at the edges of the mode-2 wave's core, and the the details of the spatial structure of the span-wise kinetic energy flux. The former could be visualized by a PIV system with a light sheet oriented in the span-wise direction, while the latter could be characterized by the more usual along-tank PIV set up. Moreover, the quantitative results of the Kelvin wave-Poincaré wave resonance and the formation of secondary Kelvin waves in our simulation should provide an easier comparison than field oriented simulations such as those of Sanchez-

Garrido and Vlasenko (2009). In terms of the numerical modeling literature, we are interested in exploring how the Schmidt number (or Prandtl number in thermally stratified systems) affects the localized shear instabilities generated near the Kelvin wave crest. This is important since Schmidt numbers representative of salt stratification ($Sc \approx 700$) are presently intractable for numerical simulations on all but the smallest scales, but realistic results may be obtained by choosing a Schmidt number larger than that for a heat stratified system ($Sc \approx 7$) but much smaller than that of salt. It also implies that while field scale simulations like those of Sanchez-Garrido and Vlasenko (2009) may have a similar Rossby number to an experimental study, they cannot have the same viscosity and diffusivity, implying that experimentalists need to carefully assess what aspects of such simulations they may successfully observe in the laboratory."

**The authors do not discuss the implications of the free-slip boundary conditions adopted in their numerical experiments; this is the case of Maxworthy (1983). For instance, how does the growth of shear instabilities near no-slip walls would change the results?**

Free slip boundaries are the commonly adopted first choice in numerical studies. No slip boundaries, and the clustered Chebyshev grids necessary to resolve them, would greatly increase the expense of the simulations. We believe that a no-slip boundary condition will have minimal impact on the Kelvin wave and on the shear instabilities near the wall, because the length scales of these instabilities are significantly larger than the boundary layer thickness. We have added a comment to the text, but carrying out no slip simulations is certainly an important future direction.

**The authors explore only one background stratification aspect ratio h1=h2, h, with different wave amplitudes. I wonder why they did not explore other background stratification. I would expect that in nature an upper layer thinner than a deep layer, or vice-versa. It might be interesting to explore how the asymmetry in layer thicknesses would change the growth and structure of K-H like instabilities.**

It is correct that nature typically has stratifications which consist of a thick deep layer and a thinner upper layer, but this break in the symmetry of layer depth leads to far more complicated behaviour (see Olsthoorn et al. (2013) and Carr et al. (2015)) coupling mode-2 and mode-1 wave generation. We decided, as many authors have in the literature on mode-2 ISWs, to begin with the simple first case of equal layer depths. We have included a comment on this in the article, and mention that unequal layer depths is a possible avenue for further investigation.

**The title of the article suggests that the focus of the manuscript is the transition to turbulence driven by second vertical mode internal waves in a rotating environment, but the paper is more general than this title. The paper examines the macro- and micro-scale processes driven by the degeneration of second vertical mode solitary waves in a rotating and bounded stratified flow. I would suggest the authors think of a different title.**

The title has been changed to "Multi-scale phenomena of rotation modified mode-2 internal waves".

**2 Specific Comments**

**1. In the introduction, one would also cite the work by Moum et al. (2003) or more recent observations by Zhang and Alford (2015), and for motivations, for instance, Cuypers et al. (2010).**

We have added references to these articles in the introduction.

**2. On page 2, between 20 and 25, I would mention the results obtained by Melville et al. (1989).**

We have done so. This was a necessary inclusion in this article.

**3. How did you estimate that a single turbulent patch has a vertical extent of about 40 cm, where are the 25 points coming from? Vertical grid points? Are the authors making a reference to Ulloa et al. (2015)'s paper?**

Yes, we are referencing Ulloa et al. (2015) in measuring their turbulent patch. We estimated that the vertical extent of the patch was 0.04 m. With a vertical grid spacing of 1.55 mm this would give a total of 25 grid points in the region. We have decided to remove the discussion on the resolution of the turbulent patch since it doesn't add anything to the current article and makes sense to revisit in future work.

**4. Subsections 1.1 and 1.2: Why equations are not enumerated? Please, enumerate them.**

We have done so.

**5. Page 3, paragraph 20: It seems obvious, but the authors could write that the domain is Lx x Ly x Lz = 6.4 m x 0.4 m x 0.3 m, or something like that.**

We have done so.

**6. Table 1: Missing units of z0 and h; both variable should be defined and schematised in Figure 1.**

The definition and units of the variables have been included. The schematic has been unchanged since the addition would cause too much clutter to the plot.

**7. Page 4, paragraph 10: ' ... we attempt to carry out a Direct Numerical Simulation (DNS) ... '. Concerning this sentence, what is the Kolmogorov scale of the numerical experiments? What is the Ozmidov scale? If authors are solving the Kolmogorov scale, please take the word 'attempt' out of the sentence, otherwise explain further.**

The text has been modified to include the following paragraph addressing this issue:

"We have run a series of direct numerical simulations (DNS) in a setup similar to that of

Maxworthy (1983), who employed a gravity intrusion from a lock release in a rotating, rectangular tank to generate mode-2 waves. Since the flow develops from a state of rest the precise definition of the term "Direct Numerical Simulation" from the turbulence literature, namely that grid spacing must be smaller than the Kolmogorov microscale, cannot be directly translated to the present situation. We define DNS in the sense commonly adopted in the stratified flow modeling literature, with Arthur and Fringer (2016) providing a concrete example. These authors state that DNS is a three-dimensional simulation which has a grid spacing which is "within approximately one order of magnitude of the Kolmogorov length scale". The Kolmogorov scale for transitional flows is defined in an ad hoc manner, usually via the explicit calculation of the viscous dissipation rate. The grid scale of our simulations is comparable to this usage since it is within an order of magnitude of the Kolmogorov scale defined from the maximum local dissipation rate. Moreover, our numerical method is spectral in all directions, and hence formally higher order than that used in Arthur and Fringer (2016). The spectral filter used to control aliasing applies only to the largest $30\%$ of wave numbers and leaves the majority untouched, and no subgrid scale model as in Large Eddy Simulation (LES) is used. Based on these considerations, and in the absence of a better term, DNS will be used throughout."

**8. Page 4, paragraph 10: Did Sanchez-Garrido and Vlasenko (2009) define in the same way the Rossby number? It seems that their Rossby number is almost twice smaller than the smaller Ro considered in this work.**

Sanchez-Garrido and Vlasenko (2009) do not define a Rossby number or Rossby radius of deformation. An estimate based on the latitude, wave speed, and channel width does give a smaller Rossby number than the one we present, however, we are not attempting to make a direct comparison to a specific geographic region. Comparison to experiments such as those carried out by Grimshaw et al. (2013) gives similar Rossby numbers.

**9. Table 2 is referenced in page 4; so far there is no clear explanation about**

**how cw and aw were estimated. The authors explained these parameters were parametrised on the wall, y = 0 m, and the amplitude was defined before the formation of instabilities. Was the emergence of interfacial instabilities defined by visual inspection of the isopycnals? How was then estimated the phase speed, cw? This matter requires a bit of further explanation.**

The initial discussion of these parameters was fairly terse, and we have now expanded their discussion and included a reference to a schematic in a previous paper. The main points (as they are now in the paper) are: $a_w$ is measured as the average maximum displacement from the upstream value of the isopycnal $\rho(z_0 \pm h)$. $c_w$ is the speed of the location of this maximum displacement. Yes, the instabilities were visually inspected from the density field.

**10. Subsection 1.2: I would suggest rewriting the description of the governing equations. I would suggest something like 'our numerical model solve the Boussinesq equations of motion on an f-plane ...'. 'stratified Navier-Stokes equations' sounds a bit unusual.**

We have done so.

**11. Page 5, paragraph 15: Is SPINS the acronym given to the numerical solver or is a general pseudo-spectral method?**

SPINS is the acronym for the numerical solver. We have fixed the typo in calling it a method as it is actually a particular solver.

**12. Page 6, paragraph 5: What type of computational resources were used to perform the numerical experiments (machine, number of cores, computational time)?**

A typical run completed in approximately 3 days on 64 processors on Compute Canada resources (SHARCNET). The higher resolution cases required significantly more resources, approximately two weeks using a combination of 64 and 128 processors

(either number was used on a restart). Data analysis requiring a day or two added additional computation time. The supercomputer cluster is mentioned in the Acknowledgments.

**13. Page 6, paragraph 5: How many vertical grid points does the numerical experiment use to solve the peaks of the square of the buoyancy frequency, N2, on the initial solitary wave? I would include this information to show that the density transitions are well solved. For instance, the vertical length-scale of the pycnocline thickness in the wavefront, h, is solved by around 17 and 36 grid points each vertical grid resolution, respectively.**

Yes, we should have discussed the resolution of the stratification. We have added the following sentences to clarify this: "For the resolution listed in table 3, the strongly stratified region of the background stratification contains approximately, $2h/\Delta z \approx 33$ points. The entire stratification has approximately 140 points, and is thus well resolved."

**14. Section 2. The authors start refereeing the work of Maxworthy (1983). Similarly to ?, I would compute the relationship between wave amplitude, Aw, phase celerity, cw, and the dependence with the controlling parameter, the Rossby number (and later with the Schmidt number). These results can be discussed along with Figure 3.**

Based on the information in table 2 we have added the following discussion: "The wave speed, measured as the speed of the location of the maximum displacement, is independent of the rotation rate and has a value larger than the linear long wave speed due to the large amplitude nature of these waves. The amplitude is only weakly dependent on the rotation rate or Rossby number."

**15. Equation 1. The definition of $\xi$ does not allow a quantitative comparison between the numerical experiments. I would recommend using a scaling that allows comparison.**

We are concerned with comparing the geometric structure of the waves between different simulations, thus the maximum-scaled KE is the best choice. We have included the maximum unscaled values in the caption for comparison.

**16. Page 6, paragraph 25. Where does the shear reach its maximum? at $\widetilde{y} = 0$?**

Yes, the shear reaches its maximum at y=0. This has been made clearer in the article.

**17. Figure 3: I would include the Rossby number for each experiment to show the background rotation environment along with the resulting wave dynamics.**

The Rossby numbers have been added to the figure caption.

**18. Section 2 (Page 7, paragraph 5) How did the authors compute the energy partition between Kelvin and Poincaré waves? The authors may compute spatial spectra as a function of time to quantify the energy contained in the spanwise and streamwise axes.**

We agree that the sentence, as written, was confusing. We have replaced it with: "Overall, the spatial distribution of kinetic energy is dominated by the Kelvin wave front and two secondary features near the focusing wall." This re-write is based on the observations made in figure 3. We note that because of the curvature of the Kelvin wave front, a spanwise spectrum will not provide an accurate representation of the energy.

**19. Page 8, paragraph 10. How is this Kelvin-Poincaré wave resonance compared with the one studied by Melville et al. (1989), and the observed by Renouard et al. (1993). It seems that the train of solitary waves obtained by Ulloa et al. (2015) also converges to azimuthal secondary Kelvin waves.**

In Renouard et al. (1993) the Poincare waves were resonantly generated as the ISW turned at a right-angled corner. We have added the following sentences:

"Renouard et al. (1986), however, matches our results of Poincare waves being resonantly generated along the side wall. Melville et al. (1989) also clearly shows that the

curvature of the wave front is due to the combination of a Poincare and Kelvin wave. Our results are in agreement with both of these results, but now applying to mode-2 ISWs."

**20. Page 9, paragraph 10: Is there any experimental evidence that a secondary Kelvin wave becomes more energetic and overtakes the leading Kelvin wave? What is the relevance of this phenomenon? This might be possible to show, for instance, by using a circular basin such as those used by Wake et al. (2005) and Ulloa et al. (2014). However, this would be a different problem. I think the authors should emphasize the relevance and implications of having secondary Kelvin waves in the system.**

We are unaware of any experimental evidence showing secondary waves surpassing the leading Kelvin wave. For rectangular geometry, this is likely to remain the case as it takes considerable time for the transfer of energy to occur. Circular basins are a different problem with additional effects due to the curvature of the side walls. We have added this point into the article.

**21. Page 12, paragraph 5: Is there any relationship between the internal Rossby radius of deformation and the length-scale where shear instabilities were found in the spanwise direction? The authors state that this region is confined to a quarter of the transversal length-scale, Lz. However, did they observe any change in terms of the Rossby radii?**

Yes, the Rossby number does play a role in determining the length-scale of the instabilities. We've added a discussion on this result.

**22. Page 12, paragraph 10: Why did the oscillatory K-H like billows disappear once the vertical resolution was increased? Any further thought? What was the most critical wavenumber in both cases, when nz = 256, and nz = 512. The authors could perform a stability analysis to understand the nature of the instabilities and the growth rate.**
For clarity, we note that the billows do not "disappear", but that small details of their evolution change. We have begun a more systematic study of even higher resolution simulations of this, and a number of other, studies in order to unravel some of the subtle ways in which hydrodynamic instability and numerical viscosity interact. Since such simulations take a long time we have cut down some of the discussion in the present manuscript. The suggestions given by the reviewer here will be taken into account in the future study. The background state is quite complex (i.e. it is not a parallel shear flow, see figure demonstrating non-negligible vertical velocity (in m/s) at focusing wall in case 10_1 below), and hence a detailed stability analysis, while a good idea, is really a separate study. (figure 1 here).

**23. Page 12, paragraph 10: Regarding the von-Karman like vortices. I would think that this kind of instabilities is possible to have a physical sense. There is a localised solitary wave propagating through a mid-layer in an initially quiescent stratified fluid. If we move on the leading solitary wave on the near-wall zone, the solitary wave feels there is a streamwise flow in the opposite direction. This scenario leads to a shear flow that could be similar than the one observed in the generation of von-Karman street vortices. Could you please give a look at the vertical velocity profile along the core of the leading wave?**

We have added the following: "These K-H billows can also be considered analogous to a von Karman vortex street. In a reference frame moving with the wave, the background flow is directed around the wave, much like it is around a cylinder in von Karman's classical experiment. The analogy is somewhat limited since the Kelvin wave core is not a solid, and due to the rotation the core bends backward away from the wall (and hence isn't a cylinder). Moreover, the instability is a shear instability, as opposed to a boundary layer separation." We have looked at the velocity, but have not included this figure in the manuscript since the density figures makes the vortices quite apparent. (Figure 2 here)

**24. Page 13, paragraph 5: Is there any clue of baroclinic-like instabilities? Have**

**the authors observed the vorticity field in the x -y plane?**

We have looked at the vorticity field and find no evidence of baroclinic-like instabilities. We have also done a back-of-the-envelope calculation for the fastest growing baroclinic mode and find that the scale of the waves in the manuscript is smaller than typical baroclinic disturbances by more than an order of magnitude.

**25. Page 14, paragraph 10: Did the authors compute the local gradient Richardson numbers on the leading solitary wave zone?**

Yes, we did. We have added the following sentence: "Though the Richardson number is not applicable in the upper and lower layers where there is no vertical density variation, within the wave core the Richardson number drops below 1/4 around the edge of the mode-2 bulge while the K-H billows form."

**26. Page 16, paragraph 10: Do the authors think that the channel width plays any role in the train of solitary-type waves structure and the shear instabilities growth? If so, please comment.**

For the instability of the leading Kelvin wave, the highest rotation rate case shows that the instability occurs within 10 cm of the focusing wall (approximately a quarter of the channel width used). Thus in fact a narrower tank would give similar results. Since the trailing Poincare and Kelvin waves are a result of the leading wave we expect them to have no impact on the shear instabilities of the leading wave. The width of the channel will, however, adjust the reflection of the Poincare waves and thus the resonance of the secondary Kelvin wave. This last point has been included.

**27. Page 17, paragraph 10: What grid resolution would be required to solve the Batchelor scale in a numerical experiment with Sc=700? How far are we to solve this problem?**

Our Kolmogorov scale (as defined from the maximum local dissipation rate) is approximately 2.5e-4 m. The Batchelor scale is therefore approximately 1e-5 m for Sc=700.

For our domain size (6.4x0.4x0.3) this would require about 7.5e14 total points. At our highest resolution we use about 5e8 total points. Hence, we are still a very long way from resolving the Sc=700 case.

**28. Page 18, paragraph 10: 'Since the stratification has broadened in this case, the Richardson number has increased leaving the unstable region for shear instabilities to form.' Please, explain better this sentence.**

The meaning in this was certainly unclear. We have changed it to read: "The higher diffusivity in the $Sc = 1$ case acts to hinder the production of shear instabilities by quickly diffusing them as they form."

**29. Results: Schmidt number dependence. What is the message from this section? In a thermally stratified fluid (Pr=7) we could expect dynamics such as those shown in Figure 14(a,b).**

We were not as clear as we could have been in the purpose of the Schmidt number dependence section. The following sentences have been included in the introduction to motivate this better.

"In terms of the numerical modeling literature, we are interested in exploring how the Schmidt number (or Prandtl number in thermally stratified systems) affects the localized shear instabilities generated near the Kelvin wave crest. This is important since Schmidt numbers representative of salt stratification (Sc $\approx 700$) are presently intractable for numerical simulations on all but the smallest scales, but realistic results may be obtained by choosing a Schmidt number larger than that for a heat stratified system (Sc $\approx 7$) but much smaller than that of salt. It also implies that while field scale simulations like those of Sanchez-Garrido and Vlasenko (2009) may have a similar Rossby number to an experimental study, they cannot have the same viscosity and diffusivity, implying that experimentalists need to carefully assess what aspects of such simulations they may successfully observe in the laboratory."

**30. There is a problem with the references, the doi link is duplicated.**

Unfortunately this is a problem with the Copernicus bibliography style. We have done what we can to fix it.

**31. Abstract: I do not understand what does the last sentence mean: 'Comparisons of equivalent cases with different Schmidt numbers indicate that while low Schmidt number results in the correct general characteristics of the modified ISWs, it does not correctly predict the trailing Poincaré wave field or the intensity and duration of the K-H instabilities'. What is not possible to predict at low Schmidt number (guessing the authors refer to Sc = 1)? How are the authors predicting the trailing Poincaré wave field and the intensity and duration of K-H instabilities?**

We have re-written the abstract to clarify this point. We had unwisely used the word 'predict' when 'model' would have been a better choice. By this we mean that a low Schmidt number does not accurately model what we have observed in the higher Schmidt number cases.

[Figure]

**Fig. 1.** Velocity along location of maximum amplitude on the focusing wall (x=1.05m in secondary figure)

[Figure]

**Fig. 2.** Vertical velocity along focusing wall during K-H billow formation. Contours are select isopycnals.